# Water induced sediment levitation enhances downslope transport on Mars

Jan Raack [1], Susan J. Conway [2], Clémence Herny[3], Matthew R. Balme[1], Sabrina Carpy[2] & Manish R. Patel [1,4]

On Mars, locally warm surface temperatures (~293 K) occur, leading to the possibility of (transient) liquid water on the surface. However, water exposed to the martian atmosphere will boil, and the sediment transport capacity of such unstable water is not well understood. Here, we present laboratory studies of a newly recognized transport mechanism: "levitation" of saturated sediment bodies on a cushion of vapor released by boiling. Sediment transport where this mechanism is active is about nine times greater than without this effect, reducing the amount of water required to transport comparable sediment volumes by nearly an order of magnitude. Our calculations show that the effect of levitation could persist up to ~48 times longer under reduced martian gravity. Sediment levitation must therefore be considered when evaluating the formation of recent and present-day martian mass wasting features, as much less water may be required to form such features than previously thought.

[1] School of Physical Sciences, Faculty of Science, Technology, Engineering & Mathematics, The Open University, Walton Hall, Milton Keynes MK7 6AA, UK. [2] Laboratoire de Planétologie et Géodynamique—UMR CNRS 6112, Université de Nantes, 2 rue de la Houssinière—BP 92208, 44322 Nantes Cedex 3, France. [3] Physikalisches Institut, Universität Bern, Sidlerstrasse 5, 3012 Bern, Switzerland. [4] Space Science and Technology Department, STFC Rutherford Appleton Laboratory, Harwell Campus, Didcot OX11 0QX, UK. Correspondence and requests for materials should be addressed to J.R. (email: jan.raack@open.ac.uk)

Downslope sediment transport can occur by dry granular flow, or alternatively can be supported by a fluid, e.g., a gas and/or a liquid. The physical properties of the interstitial fluid determines the flow behavior, which in turn influences the transport capacity of the flow[1, 2] and its final morphology. In planetary science it is extremely rare to catch sediment transport "in action" and therefore the final morphology and morphometry of the flow, often in conjunction with terrestrial analogs, are used to infer the process and the supporting fluid. On Mars, this line of reasoning has been used to infer that gullies are created by the action of liquid water[3–7] acting over timescales of potentially millions of years[4, 8–10]. One of the proposed sources of water to form these gullies comprises shallow[11] or deep aquifers[12]. Aquifer-based hypotheses have not been favored because gullies have been identified on isolated highs where groundwater is less likely to occur[3, 13–17]. Another proposed source is the melting of snow or ice under current climatic conditions[18] or in the recent past[3, 10, 19]. Non-water hypotheses include: CO$_2$-sublimation gas supported flows[20] and dry granular flows[21]. Due to their occurrence in different climatic regions (from polar regions to mid-latitudes), their different morphologies, and their different ages gullies could be formed by a variety and/or combinations of different mechanisms and no above-mentioned proposed process has yet been completely ruled out.

The present-day activity of gullies was first detected in the form of appearance of low relief, digitate, light toned deposits[22]. More recent observations include: incision of channels, formation of deposits with meter-scale relief[23–25], and dark sediment deposits within existing gullies[24, 26]. On sand dunes ongoing formation and growth of both classic and linear gullies[17, 27] as well as the seasonal occurrence of dark flows[28, 29] have been observed[27, 30, 31]. Often, but not always, found in association with gullies are dark recurring slope lineae (RSL)[32], which are characterized by their annual (re)appearance, seasonal growth during peak annual temperatures, and fading in the colder months[32–34]. These present-day surface activities have been linked to several different formation mechanisms, including liquid water (e.g., overland flow or debris flow)[17, 22, 34, 35], CO$_2$ frost sublimation and sediment fluidization[23, 26, 30, 31], liquid "cryobrines", acting in a similar way to liquid water[32, 36], or dry avalanches[37, 38]. However, we can only distinguish between these different hypotheses if we understand their associated sediment transport processes, and so we need to understand whether flows animated by liquid water behave in a similar fashion on Mars as they do on Earth. This may not be the case, because liquid water is unstable[34] under modern martian conditions.

Previous experimental work has shown that transient water under freezing conditions behaves slightly differently to stable water on Earth[39, 40], and under warmer conditions this difference is exaggerated further[41]. Remote sensing and climate models have shown that maximum surface temperatures on Mars up to ~300 K can occur during summer on Mars at equatorial- and southern mid-latitudes[32, 42], and even in the south polar regions maximum surface temperatures up to ~280 K are possible[26] meaning that transient liquid water is a possibility. As an example, detailed surface temperature analysis show that RSL only lengthen when temperatures exceed 273 K[34]. Ojha et al.[33] reported mid-afternoon maximum surface temperatures between 252 and 290 K from the Thermal Emission Imaging System at active RSL sites. Investigations with the Thermal Emission Spectrometer on RSL sites during same solar longitudes have shown maximum surface temperatures of ~296–298 K[35]. Based on these data sets, we chose two surface temperatures to investigate the contribution of transient water to downslope transport under martian environmental conditions: flows onto "cold" sediment (~278 K), and flows onto "warm" sediment (~297 K).

Our experiments reveal for the first time a transport mechanism of wet sediment levitation that occurs under low atmospheric pressures but not at terrestrial pressures. This sediment levitation effect is caused by boiling of transient water, comparable to the Leidenfrost effect, and itself triggers further sediment movement by grain avalanches. These transportation mechanisms enhance the volume of transported sediment by up to nine times and therefore reduce the required amount of liquid water to ~11% of that needed to transport the same volume of sediment without the levitation effect. Numerical scaling for gravity suggests this effect is greater for lower gravity, leading to an even greater sediment transport potential on Mars. Hence, the effect of levitation can have a direct influence on the estimated water budget for recent and present-day mass wasting processes on Mars in that the amount of water needed to transport sediment could be much smaller than previously thought.

## Results

**Experimental setup.** The experimental apparatus comprises a 0.9 × 0.4 m test-section containing a 5 cm deep sediment bed. The test section is inclined at 25° and located inside the Open University's Mars simulation chamber[40, 41] maintained at an average pressure of ~9 mbar. For each experiment, pure water was introduced near the top of the slope at 1.5 cm above the sediment bed and the resulting flow behavior was observed. The water was pumped into the chamber from an external reservoir allowing the temperature to be maintained at ~278 K and the flow rate at ~11 ml s$^{-1}$ (see Table 1). The sediment consisted of sand (~63–200 µm grain diameter). Each run was performed in triplicate, and all experiments were recorded with three cameras. Digital elevation models (DEMs) of the bed were created both before and after each run using multiview digital photogrammetry. Table 1 provides full details of the experimental conditions.

**Water flow experiments.** During the "cold" experiments water flowed over the surface of the sediment and also infiltrated into the sediment. Entrained sediment was transported downslope, depositing a series of lobes that migrated laterally over time, comparable to flows under terrestrial conditions[40]. The majority of the sediment was transported by overland flow of water (~98%; Fig. 1a–c, Supplementary Movies 1, 2). Boiling of the water was identified by the observation of bubbles at the surface. Occasional millimeter-sized, damp "pellets" of sediment were ejected by the boiling water as it infiltrated the bed. These ejected pellets transported negligible volumes of sediment (~2%). The majority of the sediment transported in the "cold" experiments was by overland flow, confined to a zone with maximum average width of ~9.2 cm and a downslope length of ~36.5 cm (Fig. 1b).

The volume of sediment transported during the "warm" experiments was nearly nine times greater than that during the cold experiments, and thus an increase in the sediment transport rate of the flow from ~0.13 cm$^3$ ml$^{-1}$ for "cold" experiments to ~1.18 cm$^3$ ml$^{-1}$ for "warm" experiments (Table 1, Fig. 2). Thus to transport the same volume of sediment in the "warm" experiment as the "cold", only ~11% of the volume of water is required. The increase in the volume transported for a given water volume in the "warm" experiments is caused by three processes: (1) transport of sediment by ballistic ejection of sediment and millimeter-sized sediment pellets, (2) transport of sediment by "levitation" of millimeter-sized to centimeter-sized sediment pellets with very rapid downslope transport, and (3) dry avalanches of sediment triggered by the ejected grains and levitating pellets. The combined effect of these processes accounted for about 96% of the total sediment transport, with

**Table 1 Summary of measured and controlled variables**

| | RUN | Mean values | | | Flow rate (ml s$^{-1}$) | Transported volume | | | | Total (cm$^3$) | Transport rate (cm$^3$ ml$^{-1}$) | Overland flow | | Pellet speed (mean values) (cm s$^{-1}$) |
| --- | --- | --- | --- | --- | --- | --- | --- | --- | --- | --- | --- | --- | --- | --- |
| | | Pressure (mbar) | Water temp. (K) | Surface temp. (K) | | Overland flow (cm$^3$) | Percolation (cm$^3$) | Pellets (cm$^3$) | Dry avalanches/ saltation (cm$^3$) | | | Max. length (cm) | Max. width (cm) | |
| "Cold" experiments | 1 | 8.2 ± 0.5 | 278.9 ± 0.05 | 278.5 ± 0.05 | 10.4 | 75.0 ± 7.6 | 10.6 ± 18.4 | 0.6 ± 1.0 | 0.0 | 86.2 ± 27.1 | 0.14 ± 0.04 | 38.1 | 7.6 | 11.3 |
| | 2 | 8.4 ± 0.8 | 278.5 ± 0.04 | 278.5 ± 0.12 | 11.2 | 71.1 ± 6.6 | 13.7 ± 12.8 | 0.7 ± 0.9 | 0.0 | 85.5 ± 20.3 | 0.13 ± 0.03 | 33.9 | 9.2 | 21.1 |
| | 3 | 8.8 ± 0.7 | 278.7 ± 0.02 | 278.4 ± 0.12 | 11.1 | 72.7 ± 3.2 | 6.8 ± 3.7 | 0.4 ± 0.2 | 0.0 | 79.9 ± 7.1 | 0.12 ± 0.01 | 37.6 | 10.6 | 25.4 |
| | Mean | 8.5 ± 0.4 | 278.7 ± 0.02 | 278.5 ± 0.06 | 10.9 | 72.9 ± 10.6 | 10.4 ± 22.7 | 0.6 ± 1.4 | 0.0 | 83.9 ± 34.6 | 0.13 ± 0.05 | 36.5 | 9.1 | 19.3 |
| "Warm" experiments | 4 | 9.8 ± 1.1 | 278.5 ± 0.02 | 296.5 ± 0.12 | 10.3 | 25.8 ± 3.4 | 225.9 ± 8.4 | 22.1 ± 15.2 | 292.4 ± 15.2 | 566.2 ± 25.5 | 0.91 ± 0.04 | 26.1 | 7.8 | 44.0 |
| | 5 | 8.7 ± 2.0 | 278.7 ± 0.04 | 297.4 ± 0.05 | 10.5 | 27.1 ± 3.1 | 222.2 ± 2.1 | 96.3 ± 14.4 | 461.0 ± 14.4 | 806.6 ± 9.0 | 1.28 ± 0.01 | 26.4 | 6.9 | 45.6 |
| | 6 | 10.5 ± 1.5 | 278.4 ± 0.04 | 296.4 ± 0.04 | 10.4 | 39.7 ± 3.2 | 220.7 ± 5.4 | 86.8 ± 14.0 | 499.7 ± 14.0 | 846.9 ± 21.7 | 1.36 ± 0.03 | 32 | 7.5 | 46.9 |
| | Mean | 9.7 ± 0.9 | 278.5 ± 0.02 | 296.8 ± 0.05 | 10.4 | 30.9 ± 5.6 | 222.9 ± 10.2 | 68.4 ± 25.2 | 417.7 ± 25.2 | 739.9 ± 34.7 | 1.18 ± 0.06 | 28.2 | 7.4 | 45.5 |

Pressure, water temperature, and surface temperature were averaged over the 60 s duration of water flow and presented with standard deviations (±values). Transported volume of sediment per unit defined by mapping (Fig. 1) and transport rate with error values (±values). See "Methods" for detailed description of error calculations

overland flow being only a minor component, in contrast to the "cold" experiments.

**Saltation and levitation processes**. The following sequence of events were reconstructed from the video footage: in the "warm" experiments, when the water came into contact with the sediment, boiling-induced saltation of the sediment created a continuous fountain of ejected grains until the sediment became saturated (after about 30 s; Fig. 3a–e, k, l; Supplementary Movies 3–5). In the very first seconds of the experiment, numerous saturated sediment pellets detached from the source area and rolled/slide quickly down the test bed (often to the end) with very little direct surface contact (Supplementary Movies 3, 4, 6). These pellets ranged in size from 0.5 to ~50 mm and were observed to travel at average speeds of ~46 cm s$^{-1}$. This is more than twice the speed of pellets under "cold" experiments (~19 cm s$^{-1}$; Table 1). We conclude that the pellets in the "warm" experiments partially levitate on a cushion of gas produced by boiling via a mechanism comparable to the Leidenfrost effect (Fig. 3a), which enhances their downslope velocity. The gas released at the base of these pellets causes erosion of loose dry sediment, as shown by tracks leading to isolated pellets, and by the formation of a short-lived transportation channel carved by a rapid series of levitating pellets in the first seconds of the experiment (Fig. 3b–d, i, j; Supplementary Movies 3, 6). The transient channel was approximately 5-cm wide and had a curvilinear shape. Due to the short length of the test bed and the fast material transport, this transient channel was backfilled within the first seven seconds (Supplementary Movies 3, 6).

**Grain avalanches**. In the "warm" experiments, the saltating sediment and levitating pellets triggered grain avalanches that propagated downslope (Fig. 1d–f, Supplementary Movies 3, 4). Grain avalanches and grain ejections occurred over the same time period (up to ~138 s), with some very late grain avalanches observed after 528 s for run 6. During the "cold" experiments no such movements were detectable. About 56% of all transported sediment was by these dry avalanches (Table 1, Fig. 2). The effect of sand saltation and grain flows caused by boiling liquid water was first reported by Massé et al.[41], who used a melting ice block as a water source, giving a very low water flow rate of 1–5 ml min$^{-1}$. They observed the formation of arcuate ridges caused by intergranular wet flow and the ejection of sand grains at the contact of the wet and dry sediment: these phenomena (but not the ridges) were also observed in our experiments. Saltation and flow arrested in their experiments once the water supply was removed[41]. In our "warm" experiments, though, saltation from the saturated sediment body continued for a mean

of ~78 s after the water was stopped. This implies that the sediment in our experiments was supersaturated, and percolation continued after removal of the water source. Supersaturation requires the water release rate to be faster than the infiltration rate (hence flow rates higher than those in Massé et al.[41]), suggesting this may be a limiting condition for sediment levitation.

**Liquid overland flow**. Liquid overland flow occurred in both "warm" and "cold" experiments, but only began in the "warm" runs at a mean of ~20 s into the experiments. The total downslope extension of the overland flow in the "warm" runs was ~76% (~8.4 cm shorter, Table 1) and the average width ~80% (~1.8 cm narrower, Table 1) of the "cold" experiments (Fig. 1, Supplementary Movies 2, 4). The average propagation rates were very similar (~0.74 cm s$^{-1}$ for the "warm" experiments, ~0.61 cm s$^{-1}$ for the "cold" experiments). The average volume of sediment mobilized by overland flow in the "warm" experiments was about half that in the "cold" experiments due to the shorter time for which this process was active.

**Scaling to martian gravity**. In our laboratory experiments we were unable to simulate the effect of martian gravity on these processes. Massé et al.[41] found that saltation induced by boiling is more effective under martian gravity than terrestrial gravity, resulting in three times more sediment transport. We do not repeat their calculations, but instead focus our attention on the effect of gravity of the levitation of pellets, in order to assess if sediment transport via this mechanism would be more or less efficient than observed in our experiments for otherwise similar conditions. Below we derive equations to describe the levitation force produced by the boiling gas, and then we apply these equations to understand the effect of gravitational acceleration on the levitation duration and the size of objects levitated.

We follow the reasoning and calculations of Diniega et al.[43] who considered the levitation of a sublimating $CO_2$ ice block on Earth and on Mars. We assume that the wet sand pellet can also be treated as a block with a width $D = 2R$ (m), a thickness $H$ (m), and an aspect ratio of $D/H$ lying on the dry sand test bed (Fig. 4). The temperature at the surface of the wet sand pellet is set at the temperature of evaporation of the liquid water $T_e$ for the relevant atmospheric pressure $p$. We assume that the temperature of the test bed $T_0$ exceeds the evaporation temperature. We assume that the gas escapes uniformly from the bottom of the object, perpendicular to the surface of the test bed. The object experiences two opposing forces (Fig. 4). The force $W$ due to weight of the object

$$W = g\rho_{ws}HA,  \qquad (1)$$

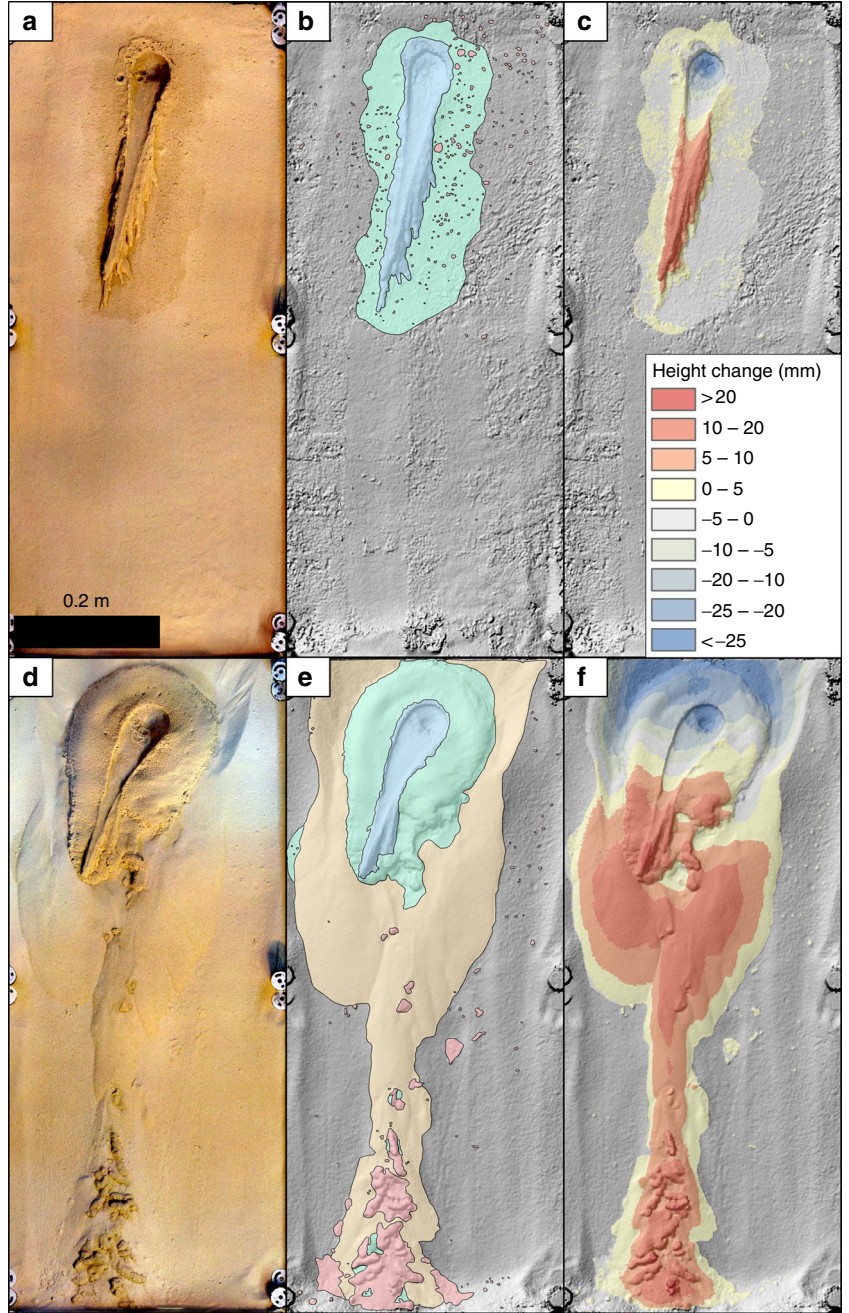

**Fig. 1** Image, map, and elevation data at the end-state of experiments. **a**, **d** Orthophotographs (0.2 mm pix$^{-1}$) of "cold" (**a**) and "warm" (**d**) experiments. **b**, **e** Hillshaded relief from DEM (1 mm pix$^{-1}$) overlain by process-zone maps giving the spatial extent of the different transport types (blue = overland flow, green = percolation, red = pellets, yellow = dry avalanches/saltation) for "cold" (**b**) and "warm" (**e**) experiments, and **c**, **f** elevation difference between start and end of "cold" (**c**) and "warm" (**f**) experiments. Flow direction is from top to bottom and the same scale is used for all images

where $g$ is the local gravity, $\rho_{ws}$ is the wet sand density, $H$ is the height of sand pellets, and $A$ is the area in contact with the test bed. As in Diniega et al.[43] we consider two shapes: (I) rectangular, if $R \ll L$ (length in m), then the problem can be solved in 2D and $A = 2RL$ or (II) cylindrical, the problem is solved in 3D and $A = \pi R^2$. The contact between the block and the sand bed results in frictional forces that prevent the block from falling. The friction force $F_T$ is proportional to the normal force $N_z$

$$F_T = \mu N_z, \qquad (2)$$

where the coefficient of proportionality $\mu$ is the Coulomb friction coefficient. Moreover, there is no motion of the pellet if

$F_T > W \sin \theta$. To determine if motion can start, we need to consider the normal force $N_z$, which is the resulting force between the weight $W$ and the levitation force $F_e$, defined in the normal direction $z$ as follows:

$$N_z = W \cos \theta - F_e, \qquad (3)$$

where $\theta$ is the slope angle and $F_e$ is the force due to the gas escape by evaporation of liquid water during boiling[43] and defined as:

$$F_e = C_f A \frac{R u_0 v}{k}. \qquad (4)$$

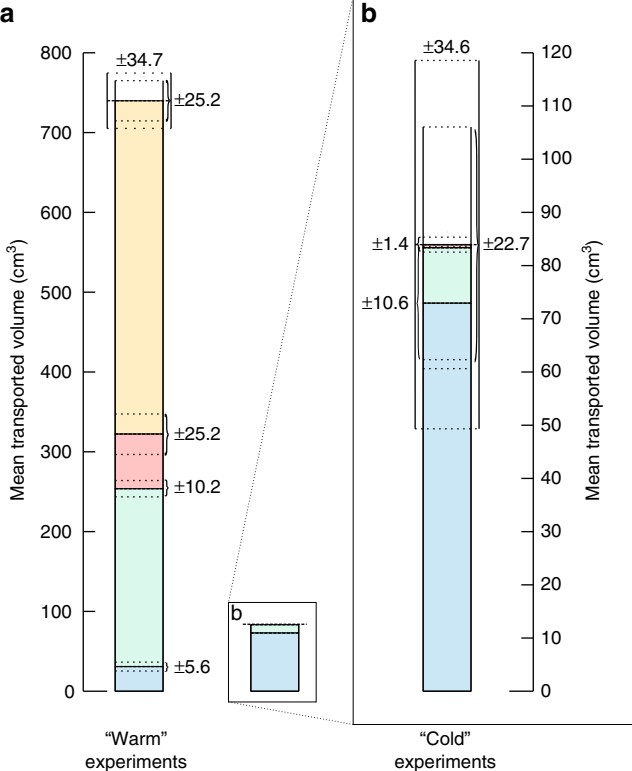

**Fig. 2** Mean volumes of transported sediment. **a** Mean volumes are divided into "warm" experiments (left bar) and "cold" experiments (right bar), and subdivided into different transport types (blue = overland flow, green = percolation, red = pellets, yellow = dry avalanches/saltation) (see also Fig. 1). The mean of total error (Measurement Error) are presented on top of the bars, errors at the side of the bars represent the mean of total errors (Measurement Error) for each individual transport type. More information on the error calculations can be found in Table 1, the methods section, and in Supplementary Table 3. **b** Re-scaled "cold" experiments

The levitation force $F_e$ is therefore proportional to the dynamic pressure $Ru_0\nu/k$ (described below), the surface area $A$, and an aerodynamic coefficient $C_f$. The aerodynamic coefficient is complex to evaluate because it depends on both the shape of the object (rectangular, spherical, oval, etc.), its roughness and the slope angle. By making some simplifying assumptions, Diniega et al.[43] have shown that this coefficient can be deduced from the calculation of the total force of the fluid exerted on the sublimating $CO_2$ block placed on a flat floor. Thus, for a rectangle ($A = 2LR$), their calculation of the force after integration gives $F_e = 2\,LR.\left(\frac{\pi}{4}\right).\left(\frac{R\,u_0\,\nu}{k}\right)$, which makes it possible to deduce that $C_f = \pi/4$ for a rectangular object and $F_e = 2LR.\left(\frac{4\pi}{3}\right).\left(\frac{R\,u_0\,\nu}{k}\right)$ is $C_f = 4/(3\pi)$ for a cylindrical object.

In our case the determination of $C_f$ is non-trivial. The pellets consist irregular objects of cohesive sand supersaturated with water. The surface of the pellets is not smooth as could be reasonably assumed for a block of $CO_2$ ice. Moreover, the shape of our objects depends on the experiment considered and can be very variable according to the temperature conditions of the experiment. Finally, we must consider the slope that will favor the levitation effect and will tend to increase this coefficient, but increasing roughness will decrease this coefficient.

For these reasons we have chosen to estimate the value of the aerodynamic coefficient $C_f$ using our experimental results. We estimated the size and shape of the pellets from the videos and orthophotos of the experiments at 297 and 278 K. The pellet sizes

range from 0.5 to 50 mm. They have irregular shapes and are often flattened with an aspect ratio $H/D = \sim 0.75$.

Therefore we know that for experiments at a sediment temperature of 297 K, the boiling effect is strong enough to move centimeter-sized pellets for the duration of several seconds. At 278 K, centimeter-sized pellets are not levitated while millimeter-sized pellets are observed to levitate for a few seconds. We tuned the aerodynamic coefficient to match these experimental observations. We find that $C_f$ ranges from approximately 1.45 to 7.3. We then used the corresponding value of the aerodynamic coefficient $C_f$ in our calculations for Mars to evaluate the influence of Mars' reduced gravity on pellet levitation.

The dynamic pressure $Ru_0\nu/k$ is dependent of the length $R$, the sand permeability $k$, the gas viscosity $\nu$, and the mean gas velocity $u_0$ escaping from the surface $A$ of the block, which is defined as follows:

$$u_0 = \frac{q}{E_v\,\rho_g}, \qquad (5)$$

where $q$ (W m$^{-2}$) is the heat flux by thermal conduction, $E_v$ is the enthalpy of evaporation for water, and $\rho_g$ is the volatile gas density. The heat flow is obtained by solving the heat equation[44]. The integration of the solution gives us the heat flux $q$ from the sand bed to the block by conduction

$$q(t) = \lambda \frac{\partial T}{\partial z}\bigg|_{z=0} = (T_0 - T_e)\sqrt{\frac{\lambda C_p\rho_s}{\pi t}}, \qquad (6)$$

where $\lambda$ is the thermal conductivity of the sand, $C_p$ is the heat capacity of the sand, $\rho_s$ is the sand density, and $t$ is the time (s).

Along a slope, a block will move if the friction force is overcome by the weight force in the $x$-direction (Fig. 4):

$$\mu(W\cos\theta - F_e) < W\sin\theta \rightarrow \tan\theta > \mu - \frac{\mu F_e}{W\cos\theta}. \qquad (7)$$

Determination of the Coulomb friction coefficient $\mu$ is non-trivial. There is no empirical method to determine this coefficient and we have no experimental measurements that allow us to calculate it directly. As $\mu$ is a coefficient, its sign is imposed, so the sense of the inequality depends only on the sign of $\left(1 - \frac{F_e}{W\cos\theta}\right)$ and therefore on the ratio $F_e/W\cos\theta$. Under the angle of repose of the sediment, if $F_e > W\cos\theta$ then the pellets will move. Increasing slope will tend to reduce the threshold value $F_T$ required to start movement.

We calculated the evolution of the ratio of the levitation force $F_e$ to the weight force $W\cos\theta$ of a block over a slope of 25° with time for both the low pressure environment of our chamber experiments at different temperatures and for equivalent conditions, but using martian gravity. The parameters used for these calculations are presented in the Supplementary Table 1. We assume that not all the heat flux due to conduction is used for the change of state of the water contains in pellets. Figure 5 shows that the levitation force produced by boiling is about 4.6 higher at 297 K than 278 K, which is consistent with our experimental results. For pellets with an aspect ratio of 0.75, circular basal area (results are similar for a rectangular base), and a sand temperature of 278 K, our calculations predict that levitation should not occur for pellets of $R = 1$ cm and should persist for about 2 s for $R = 0.1$ cm, and these predictions are consistent with our experimental observations (Supplementary Movies 1, 2). For the same aspect ratio, at sediment temperature of 297 K, our calculations predict that levitation should persist 5 s for $R = 1$ cm,

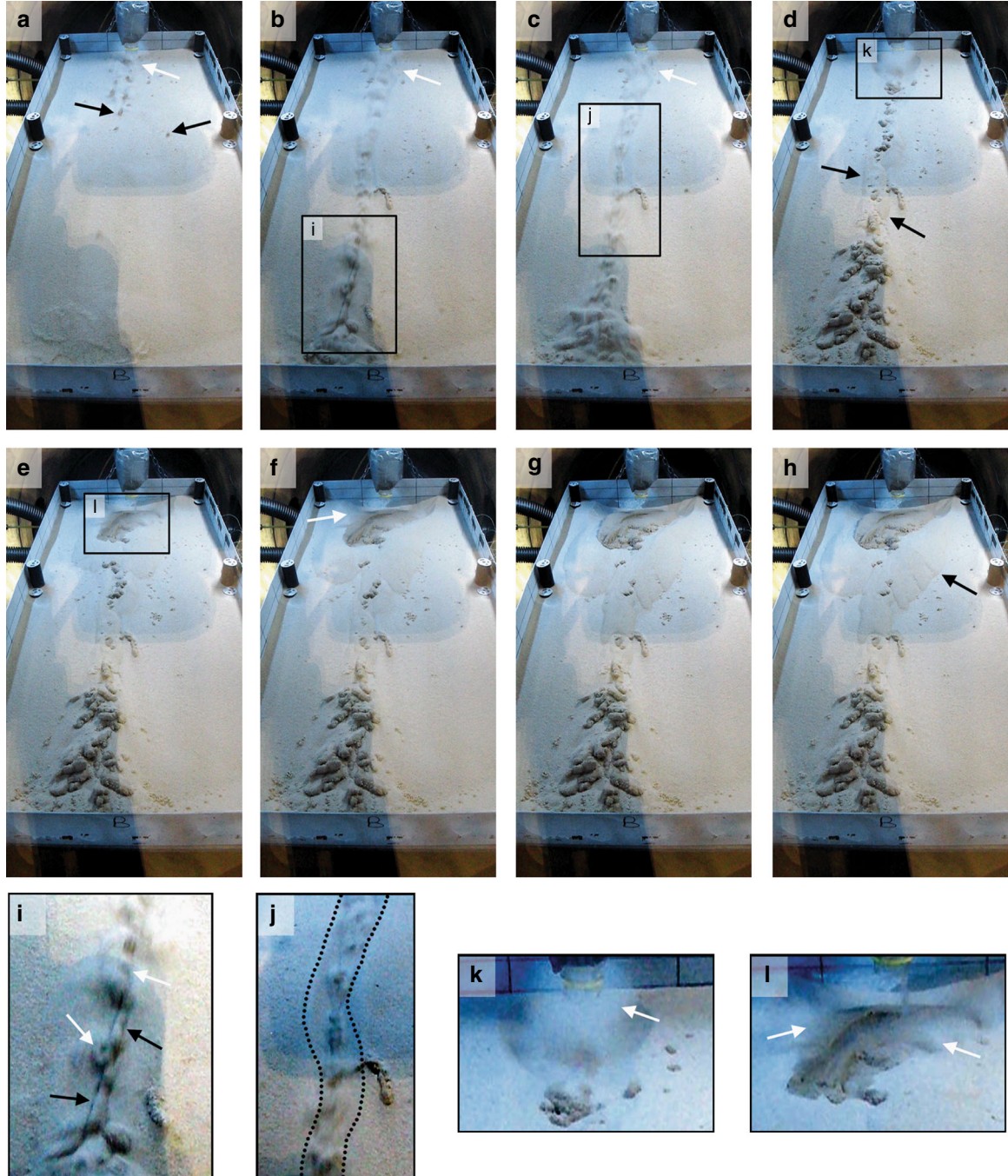

**Fig. 3** Example of transport processes. Frames from video of a "warm" temperature experiment (Run 5). Images after **a** 1 s, **b** 3 s, **c** 4 s, **d** 10 s, **e** 44 s, **f** 60 s, **g** 122 s, and **h** 303 s after the start of the experiment. White arrows point to sand saltation plumes, black arrows point to levitated pellets of wet sediment (**a**, **i**), to dry material superposing the channel (**d**), and to the last observed dry avalanche (**h**). **i–l** Show detailed excerpts of **b–e**, respectively. Contrast and brightness was adjusted on all images individually for clarity. Note for scale that the metallic tray is 0.9-m long and 0.4-m wide

which is of the same order of magnitude as the levitation duration observed in our experiments (Supplementary Movies 3, 4, 6). For $R = 0.1$ cm, we predict that levitation can last for 51.5 s, but many pellets of this size hit the end of the tray, therefore this prediction cannot be validated by our experimental data.

In our calculations we found that the levitation force produced by boiling is about 6.8 times stronger on Mars than it is in our simulation experiments (Fig. 5b). This results in an increased duration of the levitation and the possibility for larger pellets to be transported than under terrestrial gravity, even at a relatively

low surface temperature of 278 K. Therefore, the reduction of gravitational acceleration acts in favor of the levitation of pellets as well as any mass wasting triggered by the boiling of transient water[41]. Similar equations applied to levitating of $CO_2$ blocks over sand have also shown that the levitation processes is less intense on Earth than on Mars and further that a denser atmosphere also tends to inhibit levitation[43]. The temperature of the sediment plays an important role in the physics of boiling because it sets the temperature gradient between the surface and the object, which drives the heat flux powering the levitation[45].

## Discussion

Scaling to the lower martian gravity has revealed several important differences with respect to our experimental results: (1) for any given temperature condition larger sediment pellets should be levitated, (2) pellets should levitate for longer, (3) pellets should displace more sediment and create larger "channels", and (4) significant pellet-levitation should occur even under our "cold" experiment conditions.

The combination of larger pellets and the longer duration of levitation would result in a significantly larger spatial area being affected by the flow than under terrestrial gravitational acceleration. The trajectory of the majority of pellets in our "warm" experiments is interrupted by collision with the end of the test-section. A detailed reconstruction of their trajectory is beyond the scope of this work, but given the relatively high speeds of the pellets we can conservatively estimate a 2 m runout for our "warm" experiments. As the speed of the pellets is partly driven by gravitational acceleration we would expect equivalent pellets on Mars to travel more slowly (at worst at ~1/3 the speed observed in our experiments), hence their runout would likely be contained within our test-section at around 60–70 cm. Our simple calculations predict up to 48 times longer levitation of pellets considering martian gravity, which is likely to be an

overestimation, but even a levitation of 10 times longer would result in a decameter-extent of sediment disturbance, which should be visible in remote sensing images. We do not anticipate that sediment pellets themselves could reach a size readily visible in remote sensing images.

The sediment transport directly engendered by pellet levitation (excluding the secondary dry granular avalanches) is defined by the number and size of pellets that are released. The maximum size of pellets that can be levitated is defined by the levitation-force generated by boiling, yet it is likely the combination of flow-rate and infiltration rate also influences the actual sizes and numbers of pellets that are released. Because infiltration is driven by gravity, for the same quantity of water the infiltration would be 1/3 slower; however, further experiments would need to be performed to understand the exact relation between pellet size/number flow rate and infiltration rate.

To conclude, our calculations show that the levitation force is about 6.8 times stronger on Mars (Fig. 5b), resulting in levitation lasting up to 48 times longer. This would allow levitating sediments to travel decameters downslope, even with the relatively small amounts of water used in our experiments. Such disturbances could be detectable in remote sensing data, although the detailed morphologies would be unresolvable. Importantly, the calculations show that sediment levitation would be viable on Mars even under conditions similar to our "cold" experiments, so this process could be widely applicable on Mars today and in the recent past.

The driver for the enhanced transport in the "warm" runs is the combination of a rapid delivery of water to the surface and the relatively warm sediment temperature. Our experimental results do not assume a particular source of this water and below we discuss how our results might apply to the various source mechanisms already proposed. Mechanisms that deliver water rapidly to the martian surface are summarized in the context of gullies by Heldmann and Mellon[5] and Heldmann et al.[6] For example, aquifer-release[4, 6, 12, 22] is one possibility for rapid water release, but is unlikely to explain mass wasting occurring near the top of isolated dunes and massifs or crater rims[3, 15, 17, 18], although this mechanism cannot be ruled out for every gully on the surface on Mars and has recently been invoked to explain RSL[34, 46]. Our experimental results are consistent with the large range of proposed aquifer discharge rates in terms of flow rate and also the requirement of relatively warm surface temperature to melt the confining plug[47].

In the recent past in about 20% of the spin/orbital conditions (high eccentricity, high obliquity, perihelion close to the solstice) between 5 and 10 Ma ago[48, 49], the climatic conditions favor the

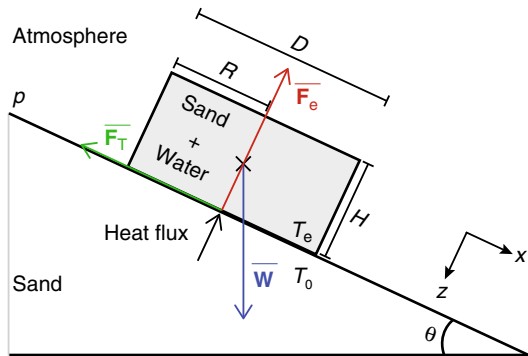

**Fig. 4** Schematic representation of a block over an inclined plain. The block represents a wet sand pellet in contact to dry sand with $T_0 > T_e$. The block is subject to three forces in competition: the weight **W**, the friction force $\mathbf{F}_T$, and the levitation force $\mathbf{F}_e$. The z-axis is oriented so that $z \geq 0$ indicates increasing depth into dry sand and the x-axis is oriented parallel to the sand surface so that absolute values of x less than 1 represent the interior of the wet sand pellet[43]

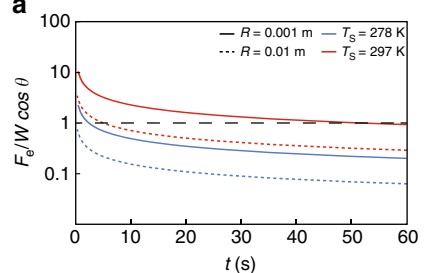
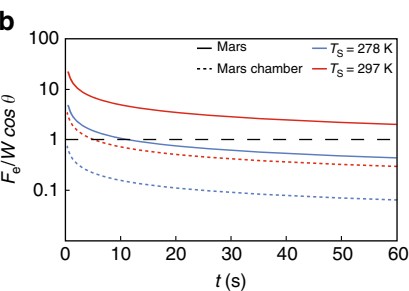

**Fig. 5** Evolution in the ratio of the levitation force to the weight force. The ratio of the levitation force $F_e$ to the weight force $W \cos \theta$ of a block over a slope of 25° with time is calculated for the physical parameters of the martian surface and of our experiments in the Mars simulation chamber (Supplementary Table 1) and use a cylindrical geometry. The levitation of the block occurs when the ratio is greater than 1 (dashed black line). **a** Ratio calculated for different sand temperatures $T_s$ and sizes of block for parameters of the Mars simulation chamber and $H/D = 0.75$. **b** Ratio calculated for different temperatures $T_s$ for parameters of martian surface and of the Mars simulation chamber and $H/D = 0.75$ with $R = 0.01$ m

formation of a near-surface ice-bearing layer that freezes and thaws regularly (e.g., daily) due to small surface temperature variations around the freezing point[50], leading to a possible availability of transient liquid water[51]. This transient liquid water could be a possible source for our observed transportation mechanism, if the parameters are correct (e.g., flow rate high enough to overcome infiltration, low pressure).

Water trapped as ice is common today as inferred from a variety of remote sensing data and modeling both within and on the martian surface[52, 53], and across a wide latitudinal range from polar[54–56] to low latitudes around ~25°[57]. For example, ice is thought to be a primary component of the mid-latitude atmospherically derived dust-ice mantle[58], as well as of mid-latitude glaciers (e.g., rock glaciers[59]). Under current climatic conditions melting would have to occur within or below isolating layers such as dust, regolith, or ice[15, 18, 60] to mitigate the effects of sublimation. Even today, locally warm surface temperatures above the frost point are possible on Mars[26, 32, 33, 35, 42]. In order for our results to be relevant such melting would have to be rapid enough to reach the flow rates in our experiments, or more likely, melt could accumulate beneath or within a protective layer before being released to the surface suddenly.

The main argument against water-based hypotheses for presently active features is they all require significant volumes of liquid water and/or brines[3, 4, 17, 22, 34–36, 46, 61]. Our newly identified process opens up the possibility that the amount of liquid water needed for present-day and geologically recent downslope transportation on Mars has been overestimated.

Furthermore, our experiments could help to answer still open questions about the formation of RSL. Although comparison with them show that their general growth-speed is relatively low (0.25–1.86 cm sol[−1])[46] compared to the sediment transportation processes observed in our experiments, there is the possibility that RSL could propagate rapidly, but episodically (e.g., only during the high peak temperatures at noon[34]), in which case their flow rates should be comparable to those used in our experiments (approximately 7 ml s[−1] as calculated from values presented by Stillman et al.[46]). The processes revealed by our experiments could also help to address the problem that RSL require a high water budget if they propagate by infiltration alone (about 1.5–5.6 m[3] m[−1] as calculated by Stillman et al.[46]). The length of RSL could be achieved with ~10 times less water (value is not scaled to martian gravity, which would make it even larger based on our calculations) if boiling is causing saltation, pellet levitation, and granular avalanches. However, our experiments concern the transport mechanisms by unstable water at the martian surface and do not inform the contentious debate about how this water was produced or brought to the surface, which is discussed in other papers[5, 6, 34, 46, 60].

The sediment transport processes we describe here are applicable anywhere in the solar system where a liquid would be unstable, and where temperature differences between the liquid and substrate could occur. Our experiments have the potential to help us understand mass wasting on bodies such as Titan[62] and Vesta[63]. However, Mars has the ideal combination of environmental parameters for this process to operate with water, as well as possessing many well-documented mass-wasting landforms that it might help explain.

The amount of water required to move sediment on martian hillslopes may have been overestimated due to the absence of the relevant sediment transport processes on Earth whose landscape is used as the basis of planetary comparison. This conclusion demonstrates the unique capability of laboratory experiments for exploring and understanding planetary surface processes, and shows how, on Mars, even a little water can go a long way.

## Methods

**Detailed experimental setup and parameter justification**. The experiments were performed in a 2-m length and 1-m wide Mars simulation chamber located at Open University. Two vacuum pumps were used to reduce the pressure at 7 mbar at which pressure all experiments were started. Due to rapid release of water vapor, average pressures for the 60 s of water flow have values around 9 mbar for each experiment. The pressure within the chamber was measured with a Pirani gauge and logged every second.

The sediment used was a natural eolian fine silica sand, with $D_{50} = 230.1$ μm and minor components of clay and silt, previously used in similar experiments at the Mars simulation chamber at the Open University[40, 41, 64]. We chose this sediment, because it is broadly consistent with sediments that are found on Mars[41, 65] and its unimodal nature aids the development of physical models. A ~5-cm depth sand bed was placed in a rectangular metallic tray (0.9-m long, 0.4-m wide, and 0.1-m deep). This thickness was chosen to avoid spill over of sediment at the end and sides of the test bed during "warm" experiments (which would influence our transport volume measurements), and to have sufficient material to avoid exposing the underlying tray upon erosion of the substrate. The angle was set to 25°, which is within the range of slope angles reported for gullies on Mars[5, 7, 16] and a compromise between the different slope angles observed at contemporary active mass wasting sites, e.g., dark flows within polar gullies (~15°)[26], linear dune gullies (~10°–20°)[17, 61], and RSL (~28°–35°)[32, 33, 35, 46]. Our chosen angle is below the angle of repose for martian and terrestrial sand dunes (between 30° and 35° based on remote sensing studies[66], and at ~30° based on experiments[67]) and hence the movements we measure are not related to dry granular flows (slip face avalanches). The water outlet was placed 1.5 cm above the sediment surface, 8 cm from the top wall of the tray. The height of the water outlet was chosen to be as close to the surface as possible, yet high enough so as not to interfere with the subsequent sediment ejection. The temperature of the sediment was monitored every second using four thermocouples placed 8 cm from the edges and 20 cm from the top/bottom at a height of ~2 cm within the sediment. Two thermocouples were used to monitor the water temperature inside the water reservoir located outside the chamber. All experiments were recorded with three different cameras, two webcams in the interior of the chamber, and one video camera outside the chamber. Each experiment was defined as a 60 s flow of water on the sediment with a water volume between 620 and 670 ml, resulting in flow rates between 10.3 and 11.2 ml s[−1]. This flow rate is intermediate between Conway et al.[40] (~80 ml s[−1]) and Massé et al.[41] (~1–5 ml s[−1]) in order to both (a) obtain erosion by overland flow under terrestrial (or non-boiling) conditions, and (b) under boiling conditions minimize the boundary effects (i.e., contact with the tray edges), but also (c) obtain a steady and reproducible flow rate. The instruments were left recording and the chamber was kept at low pressure for at least a further 10 min after the end of water flow. Averaged values and standard deviation values for pressure, water temperature, and surface temperature were calculated during the 60 s of the water flow and are presented in Table 1. Our experimental installation is somewhat comparable to that of Coleman et al.[68] with the main difference being that their experiments were performed under terrestrial pressures and not under low martian pressures, which are fundamental to observe the transport mechanism investigated in this work.

**Production of DEMs**. Before and after each experiment the sediment test bed was photographed ~40 times from a down-looking (nadir) viewpoint to construct a 3D model using "Structure-from-Motion"[69] software Agisoft PhotoScan. Twelve fixed targets of 2.67-cm diameter were positioned within the models and marked by standard black-on-white printed target markers. These targets allowed the resulting 3D models to be scaled and coregistered. Root mean square errors and reprojection errors can be found in the Supplementary Table 2. DEMs at 1 mm pix[−1] and orthophotos at 0.2 mm pix[−1] were exported to ESRI ArcGIS. The values for the volume of erosion and deposition were calculated by differencing the before and after DEMs. In order to estimate the volume transported we summed the erosion and deposition volumes and divided this number by two.

**Mapping of transport mechanisms**. In ArcGIS, all surface changes were manually mapped using the orthophotos from before and after the experiments and a hill-shaded visualization of the DEM from after the experiments. Videos were also used for additional identification of different transportation mechanisms to improve mapping. As seen in the map of Fig. 1, we used four different units for transport types: (1) overland flow (blue unit), characterized by a visible erosion and deposition of sediment via liquid water flows on top of the surface identifiable by comparing the before and after photos, inspection of the hillshade, and video observation; (2) percolation (green unit), regions where liquid water infiltrated and wetted the sediment (wetted sediment bodies) identified by the darker color in the after images, yet lack of visible transport by entrainment; (3) dry avalanches/saltation (yellow unit), characterized by the movement of dry sand (dry landslides) with no visible influence of wet sediment (no change in color); and (4) pellets (red unit), wetted sediment bodies that were ejected or detached from the source area and levitated/roll over the surface, identified by a color and elevation change, and by video observation. We used the mapped outlines to partition the volumes of sediment eroded and deposited to these different processes. This method of mapping surface changes in planview only provides a crude estimate of the

partitioning. For example, pellets deposited early in the experiments can be subsequently buried by dry flows, hence using our mapping scheme these pellets would be included in the volume assigned to the "dry avalanches/saltation" category. Therefore, we performed additional manipulations (including error calculations, described below) to improve this partitioning (Supplementary Table 3).

**Error calculations**. "Interpolation" for the overland flow in "warm" experiments. Here, we adjusted the volumes of the overland flow category calculated in planview to account for the fact that the surface was lowered by other processes. Video observations show that the area that was mapped as overland flow on the basis of the "after" DEM was heavily eroded at the beginning of the experiment by pellet ejection and dry avalanches/saltation (Supplementary Movies 3, 4). These effects are not taken into account when simply calculating volumes based on our planview mapping. Hence, we performed an adjustment by assuming the initial surface for the overland flow could be adequately approximated by an interpolated "natural neighbor" surface fitted to elevations extracted from the "after" DEM within a 2-mm buffer outside the digitized boundary (instead of using the "before" DEM as the initial surface). To quantitatively assess the uncertainty of this assumption we calculated the mean volume attributed to overland flows for "cold" experiments using the interpolation method, as described above, and compared it to the mean volume derived using the original method using the "before" DEM. We then scaled this uncertainty for the smaller area covered by the overland flow in the "warm" experiments. The "Interpolation Error" ranged between ~3.1 and ~3.4 cm$^3$ for the three "warm" experiments.

Application of the interpolation method, described above, leaves a certain volume of material unaccounted for. This volume was arbitrarily partitioned 50–50% to pellets and dry avalanches/saltation, because we do not know the exact partitioning. We consider that this 50–50 partitioning as the maximum uncertainty on the volume partitioning. The "Superposition Error" ranged from 14 to ~15 cm$^3$ for the three performed "warm" experiments.

In order to assess the "Measurement Error" associated with our volume calculations we performed test measurements on surfaces undisturbed by the flows within a fixed rectangular area (~46 cm$^2$). We made one test measurement per experiment. The resulting transport volumes were then scaled to the areas covered by particular transport types (Fig. 1b, e) to obtain the errors in their total volumes (Supplementary Table 3). If our volume calculations were perfect, the test areas should give zero-volume changes. We took this approach because uncertainty on volume calculations performed by differencing DEMs arises from a number of sources (photo quality, target misplacement, blunders in point matching, differences in lighting and texture, etc.), which are difficult to assess individually. The "Measurement Error" varied between ~1 and ~31% error for the whole area of the flows (Supplementary Table 3). For calculation of the "Total error of Runs" we scaled the "Measurement Error" to the total area. "Interpolation Error" and "Superposition Error" have no influence on the total error and only apply to the subdivision of the total volume into the different transport types (Fig. 2, Table 1, Supplementary Table 3). Mean errors for each transport type and for the "Total error of Runs" were calculated using the method of error propagation.

**Definition of overland flow**. The overland flow runoff length and width maxima were measured in ArcGIS. The runoff length is defined as the maximum linear distance between the most upslope sediment disturbance (uppermost sediment erosion via liquid water) and the lowermost sediment disturbance (lowermost sediment deposition via liquid water). The runoff width of the overland flow is defined as the maximum linear distance between the rims of the overland flow perpendicular to the runoff length. Also these values are presented in Table 1.

**Data availability**. All relevant data are available in the article and Supplementary Information files, or are available from the corresponding authors upon reasonable request.

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

## Acknowledgements

C.H. and laboratory work was funded by Europlanet (Europlanet 2020 RI has received funding from the European Union's Horizon 2020 research and innovation program under grant agreement No 654208), J.R. was funded by a Horizon 2020 Marie Skło-dowska-Curie Individual Fellowship (H2020-MSCA-IF-2014-657452), and M.R.B.'s contribution to this work was partly funded by the UK Science and Technology Facilities Council (ST/L000776/1). S.J.C. was partially supported by the French Space Agency CNES. M.R.P. was partly funded by the EU Europlanet program (grant agreement No 654208) and by the UK Science and Technology Facilities Council (ST/P001262/1). We acknowledge the help of J.P. Mason, who greatly improved our pressure measurements within the Mars simulation chamber.

## Author contributions

The methodology and experiments were planned by S.J.C. and conducted by C.H. and J.R. with significant advice and support from S.J.C. and M.R.P. Data analysis was done by J.R. and C.H. with advice from S.J.C. and M.R.B. The manuscript was prepared by J.R. with significant help and support from S.J.C., M.R.B., C.H., M.R.P., and S.J.C. Scaling calculations were done by C.H., S.C. and S.J.C.
