## [Peer Review File · Nature Communications]

Reviewers' comments:

Reviewer #1 (Remarks to the Author):

Review of Manuscript#: NCOMMS-17-07014-T; "Water induced sediment levitation enhances down-slope transport on Mars" by Raack and colleagues.

This paper presents evidence for a new sediment transport mechanism that could be unique to Mars and may help explain gullies and recurring slope lineae. Experimental results show that the temperature of the slope is an important factor with "warmer" runs transporting ~9 times more sediment than "cold" runs.

This manuscript presents intriguing results regarding several slope landforms found on Mars that are of high interest. Additionally, the documentation of the apparently unique to Mars "levitation" mechanism is an important contribution to the community. However there are some concerns with the justification for the experimental set up and how the results/methods relate to observations from Mars. This reviewer recommends publishing after minor revisions, which addresses the detailed reviewer comments below, adds some appropriate references, and fixes a figure/caption issue.

Minor issues:

1) The author's strategy for presenting a favored fluid source for Martian gullies and RSL should be clearer. It appears ice is the favorite fluid source, but this paper highlights an experiment that doesn't use an ice as its fluid source and provides counter-examples to why ice is unlikely at some sites (Pg6-7, Paragraph12). This reviewer thinks it's fine to focus on one potential fluid source, but it should fall in line with the rest of the results.

2) This paper would benefit from some discussion related to how they're experimentally derived slope morphologies may or may not be relatable to RSL and gullies on Mars. Would your results scale appropriately to RSL seen in HiRISE? Any thoughts on the rates? Judging from your experiment movies, these slope lineae form too rapidly to be analogous to the slow growth of RSL.

3) The bibliography is dominated by papers more than a decade old and largely based on older Viking and MGS orbiter data. These older papers are fine but sometimes when text is making extraordinary claims; e.g., "volumes of liquid water^{5,6,10,16,32,33}, or transport by CO₂^{12-15,32,34}." on Pg7, Paragraph13, references could provide more recent (and robust?) results.

4) Massé et al. results are appropriately discussed, but recent work by Schmidt et al. was not. This paper includes intriguing experiments that point to uniquely Mars phenomenon that are also relevant to slope processes and found RSL may be dominantly dry. Perhaps timing prohibited their inclusion but please consider discussing this paper.

Schmidt, F., F. Andrieu, F. Costard, M. Kocifaj, and A. G. Meresescu (2017), Formation of recurring slope lineae on Mars by rarefied gas-triggered granular flows, *Nature Geoscience*, 10(4), 270–273, doi:10.1038/ngeo2917.

5) Fluid composition was not clearly discussed. Was pure water used? No variations of brines were discussed as to how that might change the experimental results.

6) Line numbers would be helpful in the future.

Detailed comments:

Pg2, Paragraph1: "alternatively" might be a more appropriate choice than "alternately".

Pg2, Paragraph1: Malin et al. described their detections, as "mass movement" rather "fluidized

flow". Also, "propagation of recurring slope lineae" is too vague to most readers. Understanding you have limited length, some initial descriptions of martian phenomenon (including describing RSL) would be appropriate.

Pg3, Paragraph2: Why 300K? That was the higher end of the range McEwen described. Many low albedo mid-latitude and equatorial locations only show daily peak surface temperatures 250-273K. Both of these references looked at temperatures carefully for RSL sites and should be considered.

Ojha, L., A. McEwen, C. Dundas, S. Byrne, S. Mattson, J. Wray, M. Masse, and E. Schaefer (2014), HiRISE observations of Recurring Slope Lineae (RSL) during southern summer on Mars, *Icarus*, 231(0), 365–376, doi:10.1016/j.icarus.2013.12.021.

Stillman, D. E., T. I. Michaels, R. E. Grimm, and K. P. Harrison (2014), New observations of martian southern mid-latitude recurring slope lineae (RSL) imply formation by freshwater subsurface flows, *Icarus*, 233(0), 328–341, doi:10.1016/j.icarus.2014.01.017.

Pg3, Paragraph3: 25° is fairly shallow for many RSL slopes. 28°–35° is where they terminate and are most active (see refs below). Why this value was chosen and how it might change results needs to be discussed. Your methods state it was chosen because it is less than for dunes but not further justified. A coarser particle distribution would allow steeper slopes and would be collectively more analogous to the Martian case.

Chojnacki, M., A. McEwen, C. Dundas, L. Ojha, A. Urso, and S. Sutton (2016), Geologic context of recurring slope lineae in Melas and Coprates Chasmata, Mars: GEOLOGY OF MELAS AND COPRATES RSL, *Journal of Geophysical Research: Planets*, 121(7), 1204–1231, doi:10.1002/2015JE004991.

McEwen, A. S., C. M. Dundas, S. S. Mattson, A. D. Toigo, L. Ojha, J. J. Wray, M. Chojnacki, S. Byrne, S. L. Murchie, and N. Thomas (2014), Recurring slope lineae in equatorial regions of Mars, *Nature Geosci*, 7(1), 53–58, doi:10.1038/ngeo2014.

Dundas, C. M., A. S. McEwen, M. Chojnacki, M. P. Milazzo, and S. Byrne (2017), A Granular Flow Model for Recurring Slope Lineae on Mars, in 48th Lunar and Planetary Science Conference, p. Abstract #2399, Lunar and Planetary Institute, Houston.

Pg3, Paragraph3: 9 mbars -Why so high? Gully and RSL sites are often 2/3 to half this value.

Pg3, Paragraph3: Related to the slope choice, why such fine materials? This size range should be justified. This range of extra-fine to fine sand is appropriate for sand dunes, which was related to your justification for using a shallow slope, but many gullies and RSL are active on beds of coarser particles (coarse-sand to pebbles to small boulders). See prior citations that show HiRISE is able to resolve large particles (meter-scale) and thermal inertia measurements imply millimeter and greater size fractions.

Pg3, Paragraph4: "1.5 cm above the sediment bed" Why was this water delivery scenario chosen and what geologic source is intended to replicate? Precipitation or an atmospheric source is not discussed, but ice is.

Pg6, Paragraph11: "100 m-wide feature on Mars ..." why 100 m? That would way bigger than any slope lineae and most gully flows. Please justify.

Pg6-7, Paragraph12: This paragraph seemed a bit abrupt and off pace from paragraphs 11 and 13. 11 clearly makes the case for how the experiment is applicable to Mars, but here text somewhat dismisses certain sources of surface water. 13 mix a hard case for ice being the source, but the latter part of 12 describes why ice might be problematic. Suggest you cut part of this and just state that your preferred source is ground ice (if that is the case).

Related, the mention of potential aqueous slope processes in association with sand dunes is referred to (sometimes indirectly) and cited several places (REF: 12, 14, 16, 32, 34; Paragraph: 2, 12). However, it's unclear to this reviewer how appropriate this is, especially since your

justification for use of a shallow slope is to avoid comparison with sand dunes. Maybe as the results point to a hybrid of processes (saltation and overland flow) responsible for some Martian flow features, but in your model without water there is no saltation. Please clarify.

Pg7, Paragraph13: "Water trapped as ice is common today ...". Suggest you add "as inferred from a variety of remote sensing data" or similar. ref1 is from 1986 and ref2 is water-equivalent-hydrogen.

Figure 1. Please indicate the downhill direction.

Supplementary Movie: Movie files were not labeled properly. FYI.

Matthew Chojnacki

Reviewer #2 (Remarks to the Author):

Manuscript Summary:

This paper examines the behavior of liquid water on the surface of present-day Mars to further our understanding of sediment transport of liquid water flows. The paper presents laboratory studies to investigate how water and sediment flow downslope under martian conditions. The paper reports that sediment can "levitate" on vapor that is released during the boiling process which can increase the amount of sediment transported downslope. Consequently, the amount of water required to transport measured sediment volumes on Mars may be less than previously thought. The paper suggests that the process of sediment levitation may be an important factor for understanding transient water flows on Mars which form features such as gullies and recurring slope linea (RSL).

Overall Evaluation:

This paper addresses an interesting aspect of Mars science, namely present-day liquid water flows on Mars. However, the specific topic that the paper addresses is not novel enough to warrant publication in Nature Communications. The paper deals with understanding some of the details regarding how water flows downslope (we already know water has flowed on recent Mars). The laboratory study is not sufficiently described in the paper (or supplemental material) and would benefit from a longer and more thorough treatment in a more specialized planetary science journal. As currently presented, the work is not convincing and the calculations which are used to place constraints on the timing and volumes of water flow require more detailed explanation. The paper is also missing key references, particularly regarding martian gullies, and also does not reference (or acknowledge) previous similar work done to conduct laboratory studies of martian water flows in a Mars chamber to understand the formation of martian gullies and RSL.

Detailed Comments:

Page 1: "However, liquid water exposed to the present day martian atmosphere will boil, changing its transport behaviour".

Add the following reference which specifically modeled the simultaneously boiling and freezing of liquid water flowing through Mars gullies:

Heldmann, J.L., Toon, O.B., Pollard, W.H., Mellon, M.T., Pitlick, J., McKay, C.P., Andersen, D.T., 2005b. Formation of martian gullies by the action of liquid water flowing under current martian environmental conditions. *J. Geophys. Res.* 110, doi:10.1029/2004JE002261.

Page 1 (and throughout): The authors need to acknowledge and reference previous work that has also been focused on simulating liquid water flows to understand martian gullies (and RSL) in the laboratory setting. In particular, the University of Arkansas lab has several papers on the topic

where they simulated water flows with simultaneous boiling and freezing under Mars conditions, both with sediment transport and without.

Example publications include:

Coleman, K.A., Dixon, J.C., Howe, K.L., Roe, L.A., and V. Chevrier. Experimental simulation of martian gully forms. *Planet. and Space Sci.*, 57, doi: <https://doi.org/10.1016/j.pss.2008.11.002>, 2009.

Sears, D.W.G. and S. Moore. On laboratory simulation and the evaporation rate of water on Mars. *Geophys. Res. Ltrrs.*, 32, doi:10.1029/2005GL023443, 2005.

Sears, D.W.G. and J. Chittenden. On laboratory simulation and the temperature dependence of the evaporation rate of brine on Mars. *J. Geophys. Res.*, 32, doi: 10.1029/2005GL024154, 2005.

Moore, R. and D.W.G. Sears. On Laboratory Simulation and the Effect of Small Temperature Oscillations About the Freezing Point and Ice Formation on the Evaporation Rate of Water on Mars. *Astrobiology*, 6, doi:10.1089/ast.2006.6.644, 2006.

Page 1: The manuscript is missing several key references which are the seminal papers regarding the possible formation mechanisms for martian gullies. Be sure to include the below references for completeness when discussing Mars gully formation ("On Mars, this line of reasoning has been used to infer that gullies on Mars are created by the action of liquid water").

The Malin and Edgett (2000) paper is appropriate to cite here, but it is unclear why Costard et al. (2002) would be the (only) other paper referenced. Malin and Edgett (2000) was the first paper to report the martian gullies and suggest they were formed by liquid water. Costard et al. (2002) presents a possible formation mechanism (melting ground ice, one of many proposed ways to produce liquid water).

This broad sentence in the manuscript requires a more balanced set of references that are not specific to the formation mechanism (e.g., the source of the liquid water), but rather making the case that water was involved. The following two references are more general (acknowledging that liquid water was the likely cause of the gullies, and including all methods of generating the water that have been proposed in the literature):

Heldmann, J.L., Mellon, M.T., 2004. Observations of martian gullies and constraints on potential formation mechanisms. *Icarus* 168, 285–304.

Heldmann, J.L., Carlsson, E., Johansson, H., Mellon, M.T., and O.B. Toon. Observations of martian gullies and constraints on potential formation mechanisms, Part II: The northern hemisphere. *Icarus* 188, 324-344, 2007.

After making this general case for water carving the martian gullies, the manuscript can then cite the specific papers that proposed various sources of the water:

Shallow aquifer:

Mellon, M.T., Phillips, R.J., 2001. Recent gullies on Mars and the source of liquid water. *J. Geophys. Res.* 106, 23165–23179.

Deep aquifer:

Aquicludes:

Gilmore, M.S., Phillips, E.L., 2002. The role of aquicludes in the formation of the martian gullies. *Geology* 30, 1107–1110.

Melting ground ice:

Costard, F., Forget, F., Mangold, N., Peulvast, J.P., 2002. Formation of recent martian debris flows by melting of near-surface ground ice at high obliquity. *Science* 295, 110–113.

Melting snow:

Christensen, P.R., 2003. Formation of recent martian gullies through melting of extensive water-rich snow deposits. *Nature* 422, 45–48.

Deep aquifer:

Gaidos, E.J., 2001. Cryovolcanism and the recent flow of liquid water on Mars. *Icarus* 153, 218–223

Page 1: In reference to mass wasting forming the gullies, include the following reference which was the first paper to suggest this mechanism as a way to form Mars gullies:

Treiman, A.H., 2003. Geologic settings of martian gullies: Implications for their origins. *J. Geophys. Res.* 108, doi:10.1029/2002JE001900.

Page 3: The description of the experimental setup is insufficient. Various parameters are used for the experiment but are not adequately justified. For example, the manuscript does not explain why a 5 cm deep sediment bed was used and/or if this is a sufficient depth of sediment to confidently simulate Mars gully (or RSL) formation. The chamber pressure was held at 9 mbar, but this value is not justified. This pressure is above the triple point of water and is higher than the average martian pressure. The manuscript does not explain why a pressure of 9 mbar was chosen, or how this condition would affect the lab results and applicability to Mars. Having water so close to (or below) the triple point on Mars is at the crux of water stability on Mars and has serious consequences regarding its behavior on the martian surface. Indeed, this paper is predicated on the boiling of liquid water on Mars and the release of vapor to induce sediment levitation. A more thorough treatment of the effects of the ambient pressure on the lab results is warranted. Similarly, the laboratory flow rate of 11 ml/s was not adequately justified.

Page 3: The description of the experiments themselves is interesting and shows the promise of the laboratory runs. The differences between the “warm” and “cold” experiments are well described and given the details of the experiments presented, convincingly show that there are different physical processes affecting downslope transport for the two different scenarios.

Page 6: The manuscript uses the approach of Diniega et al. to suggest that the “lift force” on Mars is 6.8 times stronger than on Earth, which results in levitation lasting 48 times longer. However, given the information in the manuscript itself (which only states “our calculations” without additional explanation), and the insufficient information in the supplemental material, these conclusions are not adequately justified. Simply including a reference to a previous paper is not sufficient here since these calculations are a key basis for the main conclusions of the paper (e.g., increased sediment levitation and thus decreased amounts of water required for sediment transport on Mars). Since martian gravity cannot be simulated in the laboratory, these calculations are the basis for these numbers and the logic and actual calculations themselves need to be presented in this paper to substantiate these findings.

Page 6: When discussing aquifer release as a potential formation mechanism for martian gullies, include the reference to Mellon and Phillips (2001) which was the first paper to provide significant detail and analysis regarding this possible method of gully formation.

Page 6: When discussing aquifer release to form martian gullies, the manuscript states that aquifer release is unlikely to “explain mass wasting occurring near the top of isolated dunes and massifs or crater rims”. The paper also needs to include a statement to acknowledge that different gullies can form from different processes, and so gullies on isolated structures can form from different processes (e.g., some gullies can form from aquifers and others may not). Not all gullies need to form the same way. Multiple papers have made this case (for both Earth and Mars).

Pages 6-7: The manuscript includes a couple of sentences assessing the viability of melting snow to form gullies. The manuscript presents a cursory (and incomplete) treatment of this topic, simply

stating that melting would have to occur underneath a protective layer and would have to produce flow rates high enough to form gullies. The process of melting snow to form water on Mars (to feed gullies or not) is much more complex. Heldmann and Mellon (2004) and Heldmann et al. (2007) did a more thorough analysis of the viability of melting snow to form gullies. The authors of this manuscript should rethink the strategy of presenting the "warm" and "cold" scenarios in the paper. Instead of presenting incomplete analyses of formation mechanisms for gullies, if the topic of the source of water is kept in the manuscript, a more thorough discussion is warranted and the appropriate citations need to be included.

Page 6: Add the following McKay et al. (2013) reference to the discussion regarding the possibility of liquid water 5-10 Mya in high latitudes. McKay et al. (2013) present the case in the below paper (in more detail than in the present manuscript) that given the obliquity history of Mars, the martian Arctic latitudes experienced higher insolation 5-10 Mya and as a result had temperatures high enough to melt subsurface ice. These locations are thus some of the best places to have harbored near-surface liquid water in the relatively recent martian past.

Christopher P. McKay, Carol R. Stoker, Brian J. Glass, Arwen I. Davé, Alfonso F. Davila, Jennifer L. Heldmann, Margarita M. Marinova, Alberto G. Fairen, Richard C. Quinn, Kris A. Zacny, Gale Paulsen, Peter H. Smith, Victor Parro, Dale T. Andersen, Michael H. Hecht, Denis Lacelle, and Wayne H. Pollard. The Icebreaker Life Mission to Mars: A Search for Biomolecular Evidence for Life. *Astrobiology*. April 2013, 13(4): 334-353. doi:10.1089/ast.2012.0878.

Page 6: "It has the potential...". Be more specific – unclear what "it" refers to.

Reviewer #1 (Remarks to the Author):

Review of Manuscript#: NCOMMS-17-07014-T; "Water induced sediment levitation enhances down-slope transport on Mars" by Raack and colleagues.

This paper presents evidence for a new sediment transport mechanism that could be unique to Mars and may help explain gullies and recurring slope lineae. Experimental results show that the temperature of the slope is an important factor with "warmer" runs transporting ~9 times more sediment than "cold" runs.

This manuscript presents intriguing results regarding several slope landforms found on Mars that are of high interest. Additionally, the documentation of the apparently unique to Mars "levitation" mechanism is an important contribution to the community. However there are some concerns with the justification for the experimental set up and how the results/methods relate to observations from Mars. This reviewer recommends publishing after minor revisions, which addresses the detailed reviewer comments below, adds some appropriate references, and fixes a figure/caption issue.

- Many thanks for your very helpful and constructive review. With your comments we were able to improve our paper substantially and we agreed with nearly all of your comments. We added all peer-reviewed papers that you recommended into the new version of our manuscript. Below you can find the detailed answers for each of your particular comments.

In summary, following your comments and the expanded space of Nature Communications we have extensively reworked our manuscript including an improved and larger reference list, an expanded general introduction, more detailed discussion, and a more detailed methods-section. We also put some of the former supplementary text in the main manuscript. We extended our main manuscript from 2127 to 4459 words, and the reference list from 43 to 69 publications.

Minor issues:

1) The author's strategy for presenting a favored fluid source for Martian gullies and RSL should be clearer. It appears ice is the favorite fluid source, but this paper highlights an experiment that doesn't use an ice as its fluid source and provides counter-examples to why ice is unlikely at some sites (Pg6-7, Paragraph12). This reviewer thinks it's fine to focus on one potential fluid source, but it should fall in line with the rest of the results.

*- Thank you for this comment, it highlighted for us that we had conveyed a message that we did not mean to convey – namely we think that ice is the preferred source of the fluid for active martian surface processes. Our intention was not to focus on the source of the water (there are many other publications that deal with this issue in great detail), but to focus on the mechanisms of sediment transport once this water has been generated. We have now clarified this intention throughout the manuscript (e.g., by adding a paragraph headlined "Relevance of our experimental results to Mars" to our discussion section) and made a statement on lines 359-362 making this intention clear: **"However, our experiments concern the transport mechanisms by unstable water at the martian surface and do not inform the contentious debate about how this water was produced or brought to the surface, which is discussed in other papers^{6,10,11,47,59}."***

*We also changed the first part of our abstract to make it more clearer we do not favour a specific source for possible liquid water on Mars (lines 28-30): **"On Mars, locally warm surface temperatures***

(~293 K)¹ occur, leading to the possibility of liquid (transient) water on the surface^{2,3}, either derived from melting of surface/near-surface snow or ice^{4,5}, or from aquifers⁶.”

We also added this statement at the beginning of our discussion-section (lines 318-322): “Our experimental results do not assume a particular source of this water and below we discuss how our results might apply to the various source mechanisms already proposed. Mechanisms which deliver water rapidly to the martian surface are summarized in the context of gullies by Heldmann and Mellon¹⁰ and Heldmann et al.¹¹”

2) This paper would benefit from some discussion related to how they're experimentally derived slope morphologies may or may not be relatable to RSL and gullies on Mars. Would your results scale appropriately to RSL seen in HiRISE? Any thoughts on the rates? Judging from your experiment movies, these slope lineae form too rapidly to be analogous to the slow growth of RSL.

- On the one hand you are right and we could compare the morphology of the derived experimental mass wasting features to gullies or RSL on Mars. But on the other hand, and in our opinion, a morphological comparison would be based on too much speculation. Our experiments were designed to explore the physical mechanisms of sediment transport and as a result the resulting morphologies are centimetres to decametres in size due to the limited length of the Mars pressure chamber. The physical processes we investigated are not fully understood (from a physics point of view) as they involve transient, dynamic and out of equilibrium processes, where we are only starting to understand the influence of balance of the various rates (heat flow, flow propagation, pressure/humidity variations). Hence, scaling-up these morphologies is fraught with challenges until we can confidently identify the principal controlling factors. Furthermore, in order to compare morphology we would need to use comparable sediment grain sizes to those at RSL and gully sites, which vary drastically from site to site. In addition we would need to scale for Mars gravity, which again suffers from the challenges listed above for the spatial scaling. Therefore we want to leave morphologic comparison to specific surface features on Mars to a later date when our understanding of these processes is more mature, because there are too many uncertainties and this would lead to speculation rather than a constructive discussion.

Furthermore, you are right that our experimental mass movements are rapid compared to the slow growth of RSL. But it is possible that the RSL could form rapidly, but episodically (e.g., they are active only at the highest temperatures during the day, as discussed by Stillman et al. 2014) rather than growing via a constant slow movement. It is not possible to distinguish between these two possibilities from remote sensing data. Therefore we maintain that even though the phenomena in our experiments are relatively rapid, such processes acting episodically could explain the slow growth of RSL. We have made this clear on lines 349-359 of the text: “Furthermore, our experiments could help to answer still open questions about the formation of RSL. Although comparison with them show that their general growth-speed is relatively low (0.25 to 1.86 cm/sol)⁴⁷ compared to the sediment transportation-processes observed in our experiments, there is the possibility that RSL could propagate rapidly, but episodically (e.g., only during the high peak temperatures at noon⁶), in which case their flow rates should be comparable to those used in our experiments (approximately 7 ml/s as calculated from values presented by Stillman et al.⁴⁷). The processes revealed by our experiments could also help to address the problem that RSL require a high water budget if they propagate by infiltration alone (about 1.5-5.6 m³/m as calculated by Stillman et al.⁴⁷). The length of RSL could be achieved with ~10 times less water (value is not scaled to martian gravity, which would make it even larger based on our calculations) if boiling is causing saltation, pellet levitation and granular avalanches.”

3) The bibliography is dominated by papers more than a decade old and largely based on older Viking and MGS orbiter data. These older papers are fine but sometimes when text is making extraordinary claims; e.g., “volumes of liquid water^{5,6,10,16,32,33}, or transport by CO₂^{12-15,32,34}.” on Pg7, Paragraph13, references could provide more recent (and robust?) results.

- To address this comment, we have added some very recent citations to this particular passage (Stillman et al., 2014; 2016; Chojnacki et al., 2016). In general, we reworked the complete manuscript and added many more recent publications to our citation list.

4) Massé et al. results are appropriately discussed, but recent work by Schmidt et al. was not. This paper includes intriguing experiments that point to uniquely Mars phenomenon that are also relevant to slope processes and found RSL may be dominantly dry. Perhaps timing prohibited their inclusion but please consider discussing this paper.

Schmidt, F., F. Andrieu, F. Costard, M. Kocifaj, and A. G. Meresescu (2017), Formation of recurring slope lineae on Mars by rarefied gas-triggered granular flows, *Nature Geoscience*, 10(4), 270–273, doi:10.1038/ngeo2917.

- You are exactly right, we had not included that paper because it was published online on 20/03, while our manuscript was submitted on 21/03. We have not included an in-depth discussion of this paper because Schmidt et al. present numerical modelling (rather than experiments) of a new dry formation mechanism of RSL, and our study is not uniquely focused on RSL, but rather on water-driven mass wasting features under low pressures in general. Nevertheless, we have cited the work of Schmidt et al. in our introduction as part of possible contemporary surface activity on Mars triggered by dry processes.

5) Fluid composition was not clearly discussed. Was pure water used? No variations of brines were discussed as to how that might change the experimental results.

*- In our experiments we used pure water, and we have added the word “**pure**” to the description of the experiments to make it clearer to the reader (lines 90-92): “**For each experiment, pure water was introduced near the top of the slope at 1.5 cm above the sediment bed and the resulting flow behavior was observed.**”*

We think that discussing in detail the possible modifications to the processes we observed in our experiments if we exchange the liquid for a brine would bloat the discussion and remain speculative. Massé et al. observed very similar (but less intense) boiling-related processes for brine compared to pure water, yet for brines they also found new processes became active (which we don't think we can even start to speculate on). This is obviously an interesting question and should be answered in a separate experimental campaign and possibly in another publications (we would like to perform such experiments in the future).

6) Line numbers would be helpful in the future.

– *We totally agree with you and we have now explicitly added line numbers. We apologise for this oversight.*

Detailed comments:

Pg2, Paragraph1: “alternatively” might be a more appropriate choice than “alternately”.

– *Changed.*

Pg2, Paragraph1: Malin et al. described their detections, as “mass movement” rather “fluidized flow”. Also, “propagation of recurring slope linea” is too vague to most readers. Understanding you have limited length, some initial descriptions of martian phenomenon (including describing RSL) would be appropriate.

- *We extensively reworked the introduction and added several proposed formation mechanisms of gullies as well as of contemporary activity of gullies and RSL to the manuscript. We hope this helps the reader to get a quick overview of the numerous possible formation mechanisms and activities which could be triggered by liquid water.*

The changed/added text passage is as follows (lines 47-71): “On Mars, this line of reasoning has been used to infer that gullies are created by the action of liquid water^{4,9-12} acting over timescales of potentially millions of years^{9,13-15}. One of the proposed sources of water to form these gullies comprise shallow¹⁶ or deep aquifers¹⁷. Aquifer-based hypotheses have not been favored because gullies have been identified on isolated highs where groundwater is less likely to occur^{4,18-22}.

Another proposed source is the melting of snow or ice under current climatic conditions⁵ or in the recent past^{4,15,23}. Non-water hypotheses include: CO₂-sublimation gas supported flows²⁴, and dry granular flows²⁵. Due to their occurrence in different climatic regions (from polar regions to mid-latitudes), their different morphologies, and their different ages gullies could be formed by a variety and/or combinations of different mechanisms and no above mentioned proposed process has yet been completely ruled out.

The present-day activity of gullies was first detected in the form of appearance of low relief, digitate, light toned deposits²⁶. More recent observations include: incision of channels, formation of deposits with meter-scale relief²⁷⁻²⁹, and dark sediment deposits within existing gullies^{28,30}. On sand dunes ongoing formation and growth of both classic and linear gullies^{22,31} as well as the seasonal occurrence of dark flows^{32,33} have been observed^{28,34,35}. Often, but not always found in association with gullies are dark recurring slope lineae (RSL)³⁶ which are characterized by their annual (re)appearance, seasonal growth during peak annual temperatures and fading in the colder months^{6,36,37}. These present-day surface activities have been linked to several different formation mechanisms including liquid water (e.g., overland flow or debris flow)^{6,22,26,38}, CO₂ frost sublimation and sediment fluidization^{27,30,34,35}, liquid “cryobrine”, acting in a similar way to liquid water³⁶, or dry avalanches^{39,40}. However, we can only distinguish between these different hypotheses if we understand their associated sediment transport processes, and so we need to understand whether flows animated by liquid water behave in a similar fashion on Mars as they

do on Earth. This may not be the case, because liquid water is unstable⁶ under modern martian conditions.“

Pg3, Paragraph2: Why 300K? That was the higher end of the range McEwen described. Many low albedo mid-latitude and equatorial locations only show daily peak surface temperatures 250-273K. Both of these references looked at temperatures carefully for RSL sites and should be considered.

Ojha, L., A. McEwen, C. Dundas, S. Byrne, S. Mattson, J. Wray, M. Masse, and E. Schaefer (2014), HiRISE observations of Recurring Slope Lineae (RSL) during southern summer on Mars, *Icarus*, 231(0), 365–376, doi:10.1016/j.icarus.2013.12.021.

Stillman, D. E., T. I. Michaels, R. E. Grimm, and K. P. Harrison (2014), New observations of martian southern mid-latitude recurring slope lineae (RSL) imply formation by freshwater subsurface flows, *Icarus*, 233(0), 328–341, doi:10.1016/j.icarus.2014.01.017.

- Good point, we include these references and address this point as follows. Based on the works of Ojha et al. (2014) and Stillman et al. (2014) our choice of experimental temperatures is valid for the cases they study. Both papers show that surface temperatures above 273 K occurred at the time of RSL formation and growth (Ojha: between 240 K and 290 K during growth based on THEMIS; Stillman: 296±5 K during growth based on TES, and >273 K (298±5 K) based on THEMIS). It has to be acknowledged that we have poor knowledge of the maximum surface temperatures than can be experienced in favourably oriented microenvironments (such as at the source zones of RSL) from these remote sensing instruments, hence if a temperature of 273K is recorded by THEMIS, it is likely that within that pixel zones with much higher temperatures can be found (with those zones being of a scale relevant to RSL).

We changed the manuscript as follows (lines 74-85): “Remote sensing and climate models have shown that maximum surface temperatures on Mars up to ~300 K can occur during summer on Mars at equatorial- and southern mid-latitudes^{1,36}, and even in the south polar regions maximum surface temperatures up to ~280 K are possible³⁰ meaning that transient liquid water is a possibility. As an example, detailed surface temperature analysis show that RSL only lengthen when temperatures exceed 273 K⁶. Ojha et al.³⁷ reported mid-afternoon maximum surface temperatures between 252 to 290 K from the Thermal Emission Imaging System (THEMIS) at active RSL-sites. Investigations with the Thermal Emission Spectrometer (TES) on RSL-sites during same solar longitudes have shown maximum surface temperatures of ~296-298 K³⁸. Based on these datasets, we chose two surface temperatures to investigate the contribution of transient water to downslope transport under martian environmental conditions: (i) flows onto ‘cold’ sediment (~278 K) and (ii) flows onto ‘warm’ sediment (~297 K).”

Pg3, Paragraph3: 25° is fairly shallow for many RSL slopes. 28°–35° is where they terminate and are most active (see refs below). Why this value was chosen and how it might change results needs to be discussed. Your methods state it was chosen because it is less than for dunes but not further justified. A coarser particle distribution would allow steeper slopes and would be collectively more analogous to the Martian case.

Chojnacki, M., A. McEwen, C. Dundas, L. Ojha, A. Urso, and S. Sutton (2016), Geologic context of recurring slope lineae in Melas and Coprates Chasmata, Mars: GEOLOGY OF MELAS AND COPRATES RSL, *Journal of Geophysical Research: Planets*, 121(7), 1204–1231, doi:10.1002/2015JE004991.

McEwen, A. S., C. M. Dundas, S. S. Mattson, A. D. Toigo, L. Ojha, J. J. Wray, M. Chojnacki, S. Byrne, S. L. Murchie, and N. Thomas (2014), Recurring slope lineae in equatorial regions of Mars, *Nature Geosci*, 7(1), 53–58, doi:doi:10.1038/ngeo2014.

Dundas, C. M., A. S. McEwen, M. Chojnacki, M. P. Milazzo, and S. Byrne (2017), A Granular Flow Model for Recurring Slope Lineae on Mars, in 48th Lunar and Planetary Science Conference, p. Abstract #2399, Lunar and Planetary Institute, Houston.

- We agree with your comment that our chosen slope angle is relatively shallow for RSL sites. But our intention was for our experimental results to be applicable to the widest range of contemporary surface activity. A 25° slope slopes for gullies: e.g., contemporary activity of polar gullies (dark flows within existing gullies) were found on an average slope of only 15° (Raack et al., 2015) and dune gullies of the Russell crater dune field are active on slopes of 10° (Reiss et al., 2010). So, we decided to make a compromise between slopes angles observed for “active” gullies (could be low as 10°) and the relatively steep RSL slopes of 28°–35°.

We added the following information to the Methods-section in the manuscript (lines 598-604): **“The angle was set to 25°, which is within the range of slope angles reported for gullies on Mars^{10,12,21} and a compromise between the different slope angles observed at contemporary active mass wasting sites, e.g., dark flows within polar gullies (~15°)³⁰, linear dune gullies (~10-20°)^{22,60}, and RSL (~28-35°)^{36-38,47}. Our chosen angle is below the angle of repose for martian and terrestrial sand dunes (between 30° and 35° based on remote sensing studies⁶⁶, and at ~30° based on experiments⁶⁷) and hence the movements we measure are not related to dry granular flows (slip face avalanches).”**

Pg3, Paragraph3: 9 mbars -Why so high? Gully and RSL sites are often 2/3 to half this value.

- Based on the literature (e.g., Hess et al., 1980; Haberle et al., 2001) the pressure on Mars ranges between around 6.5 to 10 mbar, so we chose 7 mbar as the target pressure at the beginning of our experiments and the ~9 mbar values are the average values for the 60 seconds of water flow during the experiments. However, all pressures used in our experiments fall into the range of applicable pressures on Mars.

We changed and added the following information to the Methods-section in the manuscript (lines 586-589): **“Two vacuum pumps were used to reduce the pressure at 7 mbar at which pressure all experiments were started. Due to rapid release of water vapor, average pressures for the 60 seconds of water flow have values around 9 mbar for each experiment. The pressure within the chamber was measured with a Pirani gauge and logged every second.”**

Pg3, Paragraph3: Related to the slope choice, why such fine materials? This size range should be justified. This range of extra-fine to fine sand is appropriate for sand dunes, which was related to your justification for using a shallow slope, but many gullies and RSL are active of beds of coarser particles (coarse-sand to pebbles to small boulders). See prior citations that show HiRISE is able to resolve large particles (meter-scale) and thermal inertia measurements imply millimeter and greater size fractions.

- The sediment was chosen for the following reasons. Firstly, a large number of previous experimental campaigns concerning mass wasting features on Mars have been performed with the same grain size sand. This gives us the possibility of comparing our results with theirs more confidently. Secondly, as

stated before, the focus of our experiments is not 100% on RSL so we have to make a compromise and find the substrate fits with the widest range of active surface processes. Based on the widely used Wentworth grain size distribution (Wentworth, 1922) the grain sizes we used belongs to fine sand (125 - <250 μm). This is smaller than the grain sizes of most of the martian dunes (based on investigations of Edgett and Christensen, 1991), but comparable to grains sizes of Bagnold dunes in Gale Crater which are $\sim 150 \mu\text{m}$ in ripple troughs (e.g., Bridges et al., 2017; Lapotre et al., 2017; Johnson et al., 2017). On the other hand, it is proposed that some gullies were formed by the erosion of the atmospherically derived dust-ice mantle (e.g., Christensen, 2003; Bleamaster and Crown, 2005; Bridges and Lackner, 2006; Dickson and Head, 2009; Reiss et al., 2009; Aston et al., 2011; Schon and Head, 2011; Raack et al., 2012) whose grain size should be smaller (very fine?) than the grains size of martian dunes. Finally, unimodal aeolian sand is more favourable material when we come to developing the physical models of these processes.

We added the following information to the Methods-section (lines 590-594): **“The sediment used was a natural aeolian fine silica sand, with $D_{50}=230.1 \mu\text{m}$ and minor components of clay and silt, previously used in similar experiments at the Mars simulation chamber at the Open University^{42,43,64}. We chose this sediment, because it is broadly consistent with sediments that are found on Mars^{43,65} and its unimodal nature aids the development of physical models.”**

Pg3, Paragraph4: “1.5 cm above the sediment bed” Why was this water delivery scenario chosen and what geologic source is intended to replicate? Precipitation or an atmospheric source is not discussed, but ice is.

- Our decision was driven by practical considerations. We did not intend to directly replicate any given geologic source of the water (see above reply which details our intent to explore transport processes, rather than the generation of the liquid water itself). Ice melt with low flow rates was replicated by the experiments presented by Massé et al. (2016). With our experiments we wanted to undertake experiments with higher flow rates, which engender erosion under terrestrial conditions. We wanted to have the water outlet as near to the surface as possible but with heights below 1.5 cm the outlet apparatus would interfere with the boiling-induced transport processes. Therefore we chose 1.5 cm as the minimum height. We added the following passage to the Methods-section (lines 604-606): **“The water outlet was placed 1.5 cm above the sediment surface, 8 cm from the top wall of the tray. The height of the water outlet was chosen to be as close to the surface as possible, yet high enough so as not to interfere with the subsequent sediment ejection.”**

Pg6, Paragraph11: “100 m-wide feature on Mars ...” why 100 m? That would way bigger than any slope linea and most gully flows. Please justify.

- You are completely right, this was a mistake on our side. We should have used $\sim 5 \text{ m}$, which fits for activity for gullies and for RSL. Nevertheless, we have completely deleted this section from our manuscript, because this simple scaling to Mars had only included the width of the flow front and the transported volume of sediment, but other parameters such as the enhanced flow rate was not considered. Therefore we thought it better to avoid this consideration because its results are too speculative.

Pg6-7, Paragraph12: This paragraph seemed a bit abrupt and off pace from paragraphs 11 and 13. 11

clearly makes the case for how the experiment is applicable to Mars, but here text somewhat dismisses certain sources of surface water. 13 mix a hard case for ice being the source, but the latter part of 12 describes why ice might be problematic. Suggest you cut part of this and just state that your preferred source is ground ice (if that is the case).

- As stated in response to point #1, our intent was not to favour one process of water generation over another, so in the revision of the manuscript we have tried to remove this unintended meaning and the abrupt break remarked in this comment should no longer be an issue.

Related, the mention of potential aqueous slope processes in association with sand dunes is referred to (sometimes indirectly) and cited several places (REF: 12, 14, 16, 32, 34; Paragraph: 2, 12). However, it's unclear to this reviewer how appropriate this is, especially since your justification for use of a shallow slope is to avoid comparison with sand dunes. Maybe as the results point to a hybrid of processes (saltation and overland flow) responsible for some Martian flow features, but in your model without water there is no saltation. Please clarify.

- It is clear that we have not done a good job of communicating what we intended to! Our intention was not to avoid comparison with sand dunes. So, as stated above, we have added some new information about our chosen slope angle in the Methods-section and also a more extensive introduction which includes information about dune mass wasting forms (so making it clear our experiments are intended to be applicable also to these features). We have attempted to make it much clearer that our experiments describe a new suite of processes, which we show should be possible on Mars, and that this suite of processes could be feasible for several different recent and contemporary mass wasting forms.

Pg7, Paragraph13: "Water trapped as ice is common today ...". Suggest you add "as inferred from a variety of remote sensing data" or similar. ref1 is from 1986 and ref2 is water-equivalent-hydrogen.

*- Changed the passage and added some further literature (lines 335-337): **"Water trapped as ice is common today as inferred from a variety of remote sensing data and modelling both within and on the martian surface^{2,52}, and across a wide latitudinal range from polar⁵³⁻⁵⁵ to low latitudes around ~25⁵⁶."***

Figure 1. Please indicate the downhill direction.

- Done as suggested, changed the description of the Figure 1.

Supplementary Movie: Movie files were not labeled properly. FYI.

- We have not found any mistake in labelling of the movie files yet, but we will pay extra attention to the labelling when uploading all supplementary movies during the resubmission process.

Matthew Chojnacki

Reviewer #2 (Remarks to the Author):

Manuscript Summary:

This paper examines the behavior of liquid water on the surface of present-day Mars to further our understanding of sediment transport of liquid water flows. The paper presents laboratory studies to investigate how water and sediment flow downslope under martian conditions. The paper reports that sediment can “levitate” on vapor that is released during the boiling process which can increase the amount of sediment transported downslope. Consequently, the amount of water required to transport measured sediment volumes on Mars may be less than previously thought. The paper suggests that the process of sediment levitation may be an important factor for understanding transient water flows on Mars which form features such as gullies and recurring slope linea (RSL).

Overall Evaluation:

This paper addresses an interesting aspect of Mars science, namely present-day liquid water flows on Mars. However, the specific topic that the paper addresses is not novel enough to warrant publication in Nature Communications. The paper deals with understanding some of the details regarding how water flows downslope (we already know water has flowed on recent Mars). The laboratory study is not sufficiently described in the paper (or supplemental material) and would benefit from a longer and more thorough treatment in a more specialized planetary science journal. As currently presented, the work is not convincing and the calculations which are used to place constraints on the timing and volumes of water flow require more detailed explanation. The paper is also missing key references, particularly regarding martian gullies, and also does not reference (or acknowledge) previous similar work done to conduct laboratory studies of martian water flows in a Mars chamber to understand the formation of martian gullies and RSL.

- Thank you for your helpful review, we feel our manuscript improved substantially as a result of addressing your comments.

First, we have to state that in our opinion (and the opinion of the other reviewer) this work is novel and it is worth publishing in Nature Communications. We were unsure from your review as to which aspects of our work lead you to the conclusion that this work is not novel. So for clarity we would like to present here the aspects of our work that we think make it novel and a substantial contribution to our understanding not only of Mars, but of sediment transport on planetary surfaces: 1) transport of sediment via “levitation” of sediment pellets over warm sediment has never been reported before, and we demonstrate could be at work in recent and present-day mass wasting features on Mars. 2) we perform the first quantification (and scaling) of the enhanced sediment transport when this levitation process is active and further show this enhancement has been overlooked in previous studies and should be included when considering sediment transport on Mars. 3) We maintain that a 9-fold increase in sediment transport combined with a suite of new physical processes is not a “detail regarding how water flows downslope”. 4) Our scaling calculations have shown that the observed levitation process can be relevant even at temperatures closer to the water triple point under martian conditions. Such an increase in transport capacity means much less water is required to form any given feature than previous models predict. This has a direct impact on how scientists in the future calculate water budgets for active features on the martian surface (independent of the source of water considered). Therefore, we think that this work is novel enough for publication in Nature Communications, as well as important for the Mars community.

We would conclude the statement about the novelty of our work with a citation from the other reviewer: “This manuscript presents intriguing results regarding several slope landforms found on

Mars that are of high interest. Additionally, the documentation of the apparently unique to Mars “levitation” mechanism is an important contribution to the community.”

Second, we agree that the initial submission was lacking detail regarding our experimental procedure. We have now expanded the methods-section. We have not only reintegrated the supplementary material, but provide additional details (see replies below) including a justification of every experimental parameter choice. We feel the level of detail in the explanation of our calculations used to place constraints on the timing and volumes of water flow is already appropriate because we now include details formerly in the supplementary material into the main text (1983 words, lines 163-314). This reorganisation should make the logical progression of our arguments on the timing and volumes of water flow a lot clearer.

Thirdly, on the basis of comments from both reviewers we have extensively reworked our manuscript, notably including a larger reference list (citing more key gully-papers and previous laboratory work), but also an improved general introduction, more detailed discussion, and a more detailed methods-section. We also put the formerly supplementary text about our calculations to the main manuscript. Due to this we extended our main manuscript from 2127 to 4459 words, and the reference list from 43 to 69 publications. We hope to extinguish your concerns about the missing key references with this enhanced manuscript (we have put almost all of your proposed references to our citation list). Below you can find the detailed answers for each of your particular comments.

Detailed Comments:

Page 1: “However, liquid water exposed to the present day martian atmosphere will boil, changing its transport behaviour”.

Add the following reference which specifically modeled the simultaneously boiling and freezing of liquid water flowing through Mars gullies:

Heldmann, J.L., Toon, O.B., Pollard, W.H., Mellon, M.T., Pitlick, J., McKay, C.P., Andersen, D.T., 2005b. Formation of martian gullies by the action of liquid water flowing under current martian environmental conditions. *J. Geophys. Res.* 110, doi:10.1029/2004JE002261.

- Changed and added the reference.

Page 1 (and throughout): The authors need to acknowledge and reference previous work that has also been focused on simulating liquid water flows to understand martian gullies (and RSL) in the laboratory setting. In particular, the University of Arkansas lab has several papers on the topic where they simulated water flows with simultaneous boiling and freezing under Mars conditions, both with sediment transport and without.

Example publications include:

Coleman, K.A., Dixon, J.C., Howe, K.L., Roe, L.A., and V. Chevrier. Experimental simulation of martian gully forms. *Planet. and Space Sci.*, 57, doi: <https://doi.org/10.1016/j.pss.2008.11.002>, 2009.

Sears, D.W.G. and S. Moore. On laboratory simulation and the evaporation rate of water on Mars. *Geophys. Res. Ltr.*, 32, doi:10.1029/2005GL023443, 2005.

Sears, D.W.G. and J. Chittenden. On laboratory simulation and the temperature dependence of the evaporation rate of brine on Mars. *J. Geophys. Res.*, 32, doi: 10.1029/2005GL024154, 2005.

Moore, R. and D.W.G. Sears. On Laboratory Simulation and the Effect of Small Temperature Oscillations About the Freezing Point and Ice Formation on the Evaporation Rate of Water on Mars. *Astrobiology*, 6, doi:10.1089/ast.2006.6.644, 2006.

- We agree that we should make appropriate reference to previous laboratory studies that fall into the context of our work and now cite two of the four references suggested. Sears and Moore (2005), Sears and Chittenden (2005), Moore and Sears (2006), as well as Chevrier et al. (2007), Bryson et al. (2008), and Chevrier et al. (2008) investigated the static evaporation rates of water, ice, and brine on or below different sediment-types under Mars conditions. We do not think we need to provide an extensive review of these particular experiments within the limited space allowed, because (as pointed out by the reviewer) these works do not involve sediment transport, which is the main focus of our paper.

*We also have added the following sentence to the Method-section (lines 620-623): **“Our experimental installation is somewhat comparable to that of Coleman et al.⁶⁸ with the main difference being that their experiments were performed under terrestrial pressures and not under low martian pressures, which are fundamental to observe the transport mechanism investigated in this work.”***

Page 1: The manuscript is missing several key references which are the seminal papers regarding the possible formation mechanisms for martian gullies. Be sure to include the below references for completeness when discussing Mars gully formation (“On Mars, this line of reasoning has been used to infer that gullies on Mars are created by the action of liquid water”).

The Malin and Edgett (2000) paper is appropriate to cite here, but it is unclear why Costard et al. (2002) would be the (only) other paper referenced. Malin and Edgett (2000) was the first paper to report the martian gullies and suggest they were formed by liquid water. Costard et al. (2002) presents a possible formation mechanism (melting ground ice, one of many proposed ways to produce liquid water).

This broad sentence in the manuscript requires a more balanced set of references that are not specific to the formation mechanism (e.g., the source of the liquid water), but rather making the case that water was involved. The following two references are more general (acknowledging that liquid water was the likely cause of the gullies, and including all methods of generating the water that have been proposed in the literature):

Heldmann, J.L., Mellon, M.T., 2004. Observations of martian gullies and constraints on potential formation mechanisms. *Icarus* 168, 285–304.

Heldmann, J.L., Carlsson, E., Johansson, H., Mellon, M.T., and O.B. Toon. Observations of martian gullies and constraints on potential formation mechanisms, Part II: The northern hemisphere. *Icarus* 188, 324-344, 2007.

After making this general case for water carving the martian gullies, the manuscript can then cite the specific papers that proposed various sources of the water:

Shallow aquifer:

Mellon, M.T., Phillips, R.J., 2001. Recent gullies on Mars and the source of liquid water. *J. Geophys. Res.* 106, 23165–23179.

Deep aquifer:

Aquicludes:

Gilmore, M.S., Phillips, E.L., 2002. The role of aquicludes in the formation of the martian gullies. *Geology* 30, 1107–1110.

Melting ground ice:

Costard, F., Forget, F., Mangold, N., Peulvast, J.P., 2002. Formation of recent martian debris flows by melting of near-surface ground ice at high obliquity. *Science* 295, 110–113.

Melting snow:

Christensen, P.R., 2003. Formation of recent martian gullies through melting of extensive water-rich snow deposits. *Nature* 422, 45–48.

Deep aquifer:

Gaidos, E.J., 2001. Cryovolcanism and the recent flow of liquid water on Mars. *Icarus* 153, 218–223

- We thank the reviewer for this very complete set of references and have followed their suggested structure and referencing in our reworking of the introduction. We have added all of the proposed formation mechanisms of gullies as well as of contemporary activity of gullies and RSL (including the literature you proposed as well as other key papers) to the manuscript.

The changed/added text passage is as follows (lines 47-71): “On Mars, this line of reasoning has been used to infer that gullies are created by the action of liquid water^{4,9-12} acting over timescales of potentially millions of years^{9,13-15}. One of the proposed sources of water to form these gullies comprise shallow¹⁶ or deep aquifers¹⁷. Aquifer-based hypotheses have not been favored because gullies have been identified on isolated highs where groundwater is less likely to occur^{4,18-22}. Another proposed source is the melting of snow or ice under current climatic conditions⁵ or in the recent past^{4,15,23}. Non-water hypotheses include: CO₂-sublimation gas supported flows²⁴, and dry granular flows²⁵. Due to their occurrence in different climatic regions (from polar regions to mid-latitudes), their different morphologies, and their different ages gullies could be formed by a variety and/or combinations of different mechanisms and no above mentioned proposed process has yet been completely ruled out.

The present-day activity of gullies was first detected in the form of appearance of low relief, digitate, light toned deposits²⁶. More recent observations include: incision of channels, formation of deposits with meter-scale relief²⁷⁻²⁹, and dark sediment deposits within existing gullies^{28,30}. On sand dunes ongoing formation and growth of both classic and linear gullies^{22,31} as well as the seasonal occurrence of dark flows^{32,33} have been observed^{28,34,35}. Often, but not always found in association with gullies are dark recurring slope lineae (RSL)³⁶ which are characterized by their annual (re)appearance, seasonal growth during peak annual temperatures and fading in the colder months^{6,36,37}. These present-day surface activities have been linked to several different formation mechanisms including liquid water (e.g., overland flow or debris flow)^{6,22,26,38}, CO₂ frost sublimation and sediment fluidization^{27,30,34,35}, liquid “cryobrine”, acting in a similar way to liquid water³⁶, or dry avalanches^{39,40}. However, we can only distinguish between these different hypotheses if we understand their associated sediment transport processes, and so we need to understand whether flows animated by liquid water behave in a similar fashion on Mars as they do on Earth. This may not be the case, because liquid water is unstable⁶ under modern martian conditions.”

Page 1: In reference to mass wasting forming the gullies, include the following reference which was the first paper to suggest this mechanism as a way to form Mars gullies:
Treiman, A.H., 2003. Geologic settings of martian gullies: Implications for their origins. J. Geophys. Res. 108, doi:10.1029/2002JE001900.

-Changed and added the reference (see also the text passage above).

Page 3: The description of the experimental setup is insufficient. Various parameters are used for the experiment but are not adequately justified. For example, the manuscript does not explain why a 5 cm deep sediment bed was used and/or if this is a sufficient depth of sediment to confidently simulate Mars gully (or RSL) formation. The chamber pressure was held at 9 mbar, but this value is not justified. This pressure is above the triple point of water and is higher than the average martian pressure. The manuscript does not explain why a pressure of 9 mbar was chosen, or how this condition would affect the lab results and applicability to Mars. Having water so close to (or below) the triple point on Mars is at the crux of water stability on Mars and has serious consequences regarding its behavior on the martian surface. Indeed, this paper is predicated on the boiling of liquid water on Mars and the release of vapor to induce sediment levitation. A more thorough treatment of the effects of the ambient pressure on the lab results is warranted. Similarly, the laboratory flow rate of 11 ml/s was not adequately justified.

- Agreed and we have now revised the methods. We have provided a full justification of our parameter choice in response also to the comments of reviewer #1. The parameters were chosen on the basis of literature datasets (e.g., temperature), on the basis of former experiments (e.g., sediment material), on the basis of specifications of the pressure chamber (e.g., pressure), or on the basis of practicableness (e.g. sediment thickness). Below are the detailed explanations for all parameters:

*- Temperature: We chose 278 K for ‘cold’ and 297 K for ‘warm’ experiments based on literature datasets of martian maximum daytime surface temperatures in general and directly on RSL-sites. We have modified the manuscript as follows and added further information to the text (lines 74-85): **“Remote sensing and climate models have shown that maximum surface temperatures on Mars up to ~300 K can occur during summer on Mars at equatorial- and southern mid-latitudes^{1,36}, and even in the south polar regions maximum surface temperatures up to ~280 K are possible³⁰ meaning that transient liquid water is a possibility. As an example, detailed surface temperature analysis show that RSL only lengthen when temperatures exceed 273 K⁶. Ojha et al.³⁷ reported mid-afternoon maximum surface temperatures between 252 to 290 K from the Thermal Emission Imaging System (THEMIS) at active RSL-sites. Investigations with the Thermal Emission Spectrometer (TES) on RSL-sites during same solar longitudes have shown maximum surface temperatures of ~296-298 K³⁸. Based on these datasets, we chose two surface temperatures to investigate the contribution of transient water to downslope transport under martian environmental conditions: (i) flows onto ‘cold’ sediment (~278 K) and (ii) flows onto ‘warm’ sediment (~297 K).”***

*- Sediment thickness: The sediment thickness was chosen for practical reasons. We wanted to have a sufficient thickness of sediment so that the base of the tray would not influence the sediment transport. Test experiments have shown that ~5 cm thickness was sufficient so that the erosion of material did not reach the base of the tray. Conversely, we did not want to have so much sediment that we would loose sediment over the tray boundaries which would change our volume calculations of transported material. We added the following passage to the Methods-section (lines 594-598): **“A***

~5 cm depth sand bed was placed in a rectangular metallic tray (0.9 m long, 0.4 m wide and 0.1 m deep). This thickness was chosen (a) to avoid spill over of sediment at the end and sides of the test bed during ‘warm’ experiments (which would influence our transport volume measurements) and (b) to have sufficient material to avoid exposing the underlying tray upon erosion of the substrate.”

- Pressure: Based on the literature (e.g., Hess et al., 1980; Haberle et al., 2001) the pressure on Mars ranges between around 6.5 to 10 mbar, so we chose 7 mbar as the target pressure at the beginning of our experiments and the ~9 mbar values are the average values for the 60 seconds of water flow during the experiments. However, all pressures used in our experiments fall into the range of applicable pressures on Mars.

We changed and added the following information to the Methods-section in the manuscript (lines 586-589): **“Two vacuum pumps were used to reduce the pressure at 7 mbar at which pressure all experiments were started. Due to rapid release of water vapor, average pressures for the 60 seconds of water flow have values around 9 mbar for each experiment. The pressure within the chamber was measured with a Pirani gauge and logged every second.”**

- Water outlet height: The water outlet height of 1.5 cm was chosen for practical reasons. We wanted to have the water outlet as near to the surface as possible but with the outlet below 1.5 cm the outlet apparatus interfered with the boiling-induced transport processes. Therefore we chose 1.5 cm as the minimum height. We added the following passage to the Methods-section (lines 604-606): **“The water outlet was placed 1.5 cm above the sediment surface, 8 cm from the top wall of the tray. The height of the water outlet was chosen to be as close to the surface as possible, yet high enough so as not to interfere with the subsequent sediment ejection.”**

- Flow rate: We choose a flow rate intermediate between Conway et al. (2011) [$\sim 80 \text{ ml s}^{-1}$] and Massé et al. (2016) [$\sim 1\text{-}5 \text{ ml s}^{-1}$] in order to a) obtain erosion by overland flow under terrestrial (or non-boiling) conditions, b) under boiling conditions minimise the boundary effects (i.e. contact with the tray edges), c) obtain a steady and reproducible flow rate. We added the following passage to the Methods-section (lines 611-616): **“Each experiment was defined as a 60 s flow of water on the sediment with a water volume between 620 and 670 ml, resulting in flow rates between 10.3 and 11.2 ml s⁻¹. This flow rate is intermediate between Conway et al.⁴² [$\sim 80 \text{ ml s}^{-1}$] and Massé et al.⁴³ [$\sim 1\text{-}5 \text{ ml s}^{-1}$] in order to both a) obtain erosion by overland flow under terrestrial (or non-boiling) conditions, and b) under boiling conditions minimize the boundary effects (i.e. contact with the tray edges), but also c) obtain a steady and reproducible flow rate.”**

- Slope angle: As stated briefly, previously in our Methods-section the slope angle was chosen based on the literature (e.g., Heldmann and Mellon, 2004; Dickson et al., 2007, Conway et al. 2015) for gullies. We have now expanded this justification, following comments also from reviewer #1. We cite work that find that contemporary activity of polar gullies (dark flows within existing gullies) are found on slopes with an average slope of only 15° (Raack et al., 2015) and dune gullies of the Russell crater dune field are active on slopes with 10° (Reiss et al., 2010). RSL can be found on much steeper slopes (around 28-35° [McEwen et al., 2011]). We sought an intermediate slope angle, making a compromise between slopes angles with “active” gullies (could be low as 10°) and the relatively steep RSL slopes. Therefore we chose 25° which will fit all our requirements.

We added the following information to the Methods-section in the manuscript (lines 598-604): **“The angle was set to 25°, which is within the range of slope angles reported for gullies on Mars^{10,12,21} and a compromise between the different slope angles observed at contemporary active mass wasting sites, e.g., dark flows within polar gullies ($\sim 15^\circ$)³⁰, linear dune gullies ($\sim 10\text{-}20^\circ$)^{22,60}, and RSL ($\sim 28\text{-}35^\circ$)^{36-38,47}. Our chosen angle is below the angle of repose for martian and terrestrial sand dunes (between 30° and 35° based on remote sensing studies⁶⁶, and at $\sim 30^\circ$ based on**

experiments⁶⁷) and hence the movements we measure are not related to dry granular flows (slip face avalanches)."

*- Sediment material: - The sediment was chosen for the following reasons. Firstly, previous experimental campaigns concerning mass wasting features on Mars have been performed with the same grain size sand (Conway et al., 2011; Jouannic et al., 2015; Massé et al., 2016). This gives us the possibility of comparing our results with theirs more confidently. Secondly, we have to make a compromise and find the substrate fits with the widest range of active surface processes. Based on the widely used Wentworth grain size distribution (Wentworth, 1922) the grain sizes we used belongs to fine sand (125 - <250 μm). This is smaller than the grain sizes of most of the martian dunes (based on investigations of Edgett and Christensen, 1991), but comparable to grains sizes of Bagnold dunes in Gale Crater which are $\sim 150 \mu\text{m}$ in ripple troughs (e.g., Bridges et al., 2017; Lapotre et al., 2017; Johnson et al., 2017). On the other hand, it is proposed that some gullies were formed by the erosion of the atmospherically derived dust-ice mantle (e.g., Christensen, 2003; Bleamaster and Crown, 2005; Bridges and Lackner, 2006; Dickson and Head, 2009; Reiss et al., 2009; Aston et al., 2011; Schon and Head, 2011; Raack et al., 2012) whose grain size should be smaller (very fine?) than the grains size of martian dunes. Finally, unimodal aeolian sand is more favourable material when we come to developing the physical models of these processes. We added the following information to the Methods-section (lines 590-594): **"The sediment used was a natural aeolian fine silica sand, with $D_{50}=230.1 \mu\text{m}$ and minor components of clay and silt, previously used in similar experiments at the Mars simulation chamber at the Open University^{42,43,64}. We chose this sediment, because it is broadly consistent with sediments that are found on Mars^{43,65} and its unimodal nature aids the development of physical models."***

Page 3: The description of the experiments themselves is interesting and shows the promise of the laboratory runs. The differences between the "warm" and "cold" experiments are well described and given the details of the experiments presented, convincingly show that there are different physical processes affecting downslope transport for the two different scenarios.

- Thank you!

Page 6: The manuscript uses the approach of Diniega et al. to suggest that the "lift force" on Mars is 6.8 times stronger than on Earth, which results in levitation lasting 48 times longer. However, given the information in the manuscript itself (which only states "our calculations" without additional explanation), and the insufficient information in the supplemental material, these conclusions are not adequately justified. Simply including a reference to a previous paper is not sufficient here since these calculations are a key basis for the main conclusions of the paper (e.g., increased sediment levitation and thus decreased amounts of water required for sediment transport on Mars). Since martian gravity cannot be simulated in the laboratory, these calculations are the basis for these numbers and the logic and actual calculations themselves need to be presented in this paper to substantiate these findings.

We have added a more informative introduction to our scaling calculations and sub-divided their discussion into derivation and results and removed the complete section from the supplementary material to the main manuscript (lines 165-316). We have also removed one of the more speculative sections concerning the spatial scaling whose logical progression was unclear and on reflection cannot be supported strongly by our experimental or theoretical treatment.

Page 6: When discussing aquifer release as a potential formation mechanism for martian gullies, include the reference to Mellon and Phillips (2001) which was the first paper to provide significant detail and analysis regarding this possible method of gully formation.

- *Changed and added to reference. The information is now (lines 49-50): **“One of the proposed sources of water to form these gullies comprise shallow¹⁶ or deep aquifers¹⁷.”***

Page 6: When discussing aquifer release to form martian gullies, the manuscript states that aquifer release is unlikely to “explain mass wasting occurring near the top of isolated dunes and massifs or crater rims”. The paper also needs to include a statement to acknowledge that different gullies can form from different processes, and so gullies on isolated structures can form from different processes (e.g., some gullies can form from aquifers and others may not). Not all gullies need to form the same way. Multiple papers have made this case (for both Earth and Mars).

- *This is completely right and we totally agree with your comment. We have added this sentence to the introduction (lines 54-57): **“Due to their occurrence in different climatic regions (from polar regions to mid-latitudes), their different morphologies, and their different ages gullies could be formed by a variety and/or combinations of different mechanisms and no above mentioned proposed process has yet been completely ruled out.”***

*Furthermore, we added a statement at the end of the initial aquifer line (lines 322-325): **“For example, aquifer-release^{9,16,17,26} is one possibility for rapid water release, but is unlikely to explain mass wasting occurring near the top of isolated dunes and massifs or crater rims^{4,5,20,22}, although this mechanism cannot be ruled out for every gully on the surface on Mars and has recently been invoked to explain RSL^{6,47}.”***

Pages 6-7: The manuscript includes a couple of sentences assessing the viability of melting snow to form gullies. The manuscript presents a cursory (and incomplete) treatment of this topic, simply stating that melting would have to occur underneath a protective layer and would have to produce flow rates high enough to form gullies. The process of melting snow to form water on Mars (to feed gullies or not) is much more complex. Heldmann and Mellon (2004) and Heldmann et al. (2007) did a more thorough analysis of the viability of melting snow to form gullies. The authors of this manuscript should rethink the strategy of presenting the “warm” and “cold” scenarios in the paper. Instead of presenting incomplete analyses of formation mechanisms for gullies, if the topic of the source of water is kept in the manuscript, a more thorough discussion is warranted and the appropriate citations need to be included.

- *We have now made it clear (in response to this comment and those of reviewer #1) that the aim of the paper is not to assess or weigh up the different sources of water that others have proposed, but to say how our newly discovered transport mechanism contributes to the debates on those different sources (mainly influencing arguments based on water budgets). We think a detailed discussion regarding the source water for gullies goes beyond the scope of this paper. Our intention was not to focus on the source of the water (there are many other publications that deal with this issue in great detail), but to focus on the mechanisms of sediment transport once this water has been generated. We have now clarified this intention throughout the text and made a clear statement on lines 359-362, making this intention clear: **“However, our experiments concern the transport mechanisms by***

unstable water at the martian surface and do not inform the contentious debate about how this water was produced or brought to the surface, which is discussed in other papers^{6,10,11,47,59}.

So we outline the current ideas regarding possible water sources on Mars now and in the recent past that are already present in the literature. Therefore, we have made this intention more clear in the manuscript. We added the following sentence to our manuscript (lines 318-322), citing the papers mentioned: ***“Our experimental results do not assume a particular source of this water and below we discuss how our results might apply to the various source mechanisms already proposed. Mechanisms which deliver water rapidly to the martian surface are summarized in the context of gullies by Heldmann and Mellon¹⁰ and Heldmann et al.¹¹.”***

Page 6: Add the following McKay et al. (2013) reference to the discussion regarding the possibility of liquid water 5-10 Mya in high latitudes. McKay et al. (2013) present the case in the below paper (in more detail than in the present manuscript) that given the obliquity history of Mars, the martian Arctic latitudes experienced higher insolation 5-10 Mya and as a result had temperatures high enough to melt subsurface ice. These locations are thus some of the best places to have harbored near-surface liquid water in the relatively recent martian past.

Christopher P. McKay, Carol R. Stoker, Brian J. Glass, Arwen I. Davé, Alfonso F. Davila, Jennifer L. Heldmann, Margarita M. Marinova, Alberto G. Fairen, Richard C. Quinn, Kris A. Zacny, Gale Paulsen, Peter H. Smith, Victor Parro, Dale T. Andersen, Michael H. Hecht, Denis Lacelle, and Wayne H. Pollard. The Icebreaker Life Mission to Mars: A Search for Biomolecular Evidence for Life. *Astrobiology*. April 2013, 13(4): 334-353. doi:10.1089/ast.2012.0878.

- *We have added this reference to our manuscript.*

Page 6: “It has the potential....”. Be more specific – unclear what “it” refers to.

- *Changed to (lines 365-366): “Our experiments have the potential to help us understand mass wasting on bodies such as Titan⁶² and Vesta⁶³.”*

REVIEWERS' COMMENTS:

Reviewer #1 (Remarks to the Author):

Review of Manuscript#: Nature Communications manuscript NCOMMS1707014A, by Raack and colleagues.

Thank you for the revisions. I find them to be sufficient. Regarding the Supplementary Movie file name, perhaps there is a file conversion by Nature or I do not understand the etiquette, but they still seem unclear to me. For example, the file name for what I perceive is "Supplementary Movie 1" is 125298_1_video_2402524_wrwwxw.mov. There is no traceability of that file to the S1 movie or the text in general.

Reviewer #2 (Remarks to the Author):

This is a review to address the revised manuscript "Water induced sediment levitation enhances downslope transport on Mars" (NCOMMS-17-07014A).

The authors have completed a thorough revision, extending the length of the manuscript to include additional information requested by the Reviewers, and also expanding the reference list. The main issues (pointed out by both Reviewers) were 1) the lack of detail in justifying the experimental setup, and 2) the incomplete discussion of the possible formation mechanisms of the gullies and how the source of the water does (or does not, in this case) affect the experimental results. The authors have heeded the suggestions of the Reviewers and I am now satisfied with the treatment of the subject matter. Additional detail and discussion has been added to the manuscript text, and appropriate citations are now referenced. The paper is publishable in current form, although it is of course at the Editor's discretion regarding the suitability to Nature Communications.

Reviewer #1 (Remarks to the Author):

Review of Manuscript#: Nature Communications manuscript NCOMMS1707014A, by Raack and colleagues.

Thank you for the revisions. I find them to be sufficient. Regarding the Supplementary Movie file name, perhaps there is a file conversion by Nature or I do not understand the etiquette, but they still seem unclear to me. For example, the file name for what I perceive is "Supplementary Movie 1" is 125298_1_video_2402524_wrwww.mov. There is no traceability of that file to the S1 movie or the text in general.

Thank you again for your helpful review. Regarding the Supplementary Movie file names: It seems that there has been renaming during the file conversion by the online system. I originally named the Movies "Supplementary Movie 1", "Supplementary Movie 2", etc. using the same name and order as appears in the manuscript. We will look out for this type of issue for the final submission.

Reviewer #2 (Remarks to the Author):

This is a review to address the revised manuscript "Water induced sediment levitation enhances downslope transport on Mars" (NCOMMS-17-07014A).

The authors have completed a thorough revision, extending the length of the manuscript to include additional information requested by the Reviewers, and also expanding the reference list. The main issues (pointed out by both Reviewers) were 1) the lack of detail in justifying the experimental setup, and 2) the incomplete discussion of the possible formation mechanisms of the gullies and how the source of the water does (or does not, in this case) affect the experimental results. The authors have heeded the suggestions of the Reviewers and I am now satisfied with the treatment of the subject matter. Additional detail and discussion has been added to the manuscript text, and appropriate citations are now referenced. The paper is publishable in current form, although it is of course at the Editor's discretion regarding the suitability to Nature Communications.

Thank you again for your helpful review and we are pleased that the reviewer is satisfied that our manuscript has been revised sufficiently to be ready for publication.